# Cognitive impairment and dementia— Are they linked to childhood health and socioeconomic status? A systematic review

Tung Le[1]*, Asri Maharani[1‡], Mark Hayter[2], James Gilleen[1], Amanda Lee[2]

1 Mental Health Research Group, Division of Nursing, Midwifery and Social Work, Faculty of Biology, Medicine and Health, School of Health Sciences, The University of Manchester, Manchester, United Kingdom, 2 Faculty of Health and Education, School of Nursing and Public Health, Manchester Metropolitan University, Manchester, United Kingdom

☮ These authors contributed equally to this work.
‡ AM also contributed equally to this work.
* tung.le@manchester.ac.uk

## Abstract

### Background

Dementia is a major public health concern, with its incidence rising as the population ages. Recent studies suggest links between childhood health, socioeconomic status, and later-life cognitive impairment and dementia, though findings remain inconclusive. This systematic review evaluates the influence of childhood health and socioeconomic status on cognitive impairment and dementia.

### Method and findings

A systematic search conducted in MEDLINE, CiNAHL, and PsycINFO in December 2024 identified 44 studies matching our inclusion criteria. Findings are presented under five key themes: (1) childhood health, (2) childhood educational attainment, (3) family socio-economic and educational factors, (4) childhood experiences, and (5) childhood reading habits and social interactions.

### Conclusion

Our results highlight the need for further longitudinal studies to establish causal relationships between early-life risk factors and later cognitive decline. Policymakers should prioritize early childhood development programs that integrate health, nutrition, education, and social support to help mitigate cognitive impairment and dementia in later life.

## Introduction

Our aging population will pass 2 billion by 2050 [1], meaning we face urgent challenges to mitigate age-related diseases, particularly Alzheimer's disease and other dementias [2]. Dementia care has become a top public health priority due to its widespread prevalence

**Data availability statement:** All relevant data are within the paper and its Supporting Information files.

**Funding:** The author(s) received no specific funding for this work.

and lack of curative treatments [3]. Consequently, understanding causation is crucial to underpin targeted early interventions and diagnoses. It can inform policy and healthcare strategies that may mitigate the significant burden of dementia and reinforce the disease as a public health priority.

The Lancet Commission on Dementia Prevention, Intervention, and Care identified 14 modifiable risk factors, including childhood-related factors such as low education, which account for 40% of global dementia cases [3]. Lifecourse preventative strategies are supported by the World Health Organization (WHO) 2019 report, "Reducing the Risk of Cognitive Decline and Dementia" [4]. Their guidelines highlight the impact of early-life interventions in mitigating cognitive decline and reinforce a need to evaluate childhood health and socioeconomic conditions as potential contributors to later-life cognitive decline. Thus, this systematic review seeks to reveal early-life risk factors linked to cognitive impairment and dementia.

Malnutrition affects over 149 million children under five and has significant long-term neurodevelopmental consequences [5]. Over 356 million children are living in extreme poverty and lacking access to basic healthcare, food, education, and resources [6]. There is evidence linking poor health or adverse socioeconomic conditions in childhood with later-stage cognitive decline, suggesting those from poorer backgrounds had a 1.5 to 2.0 times higher risk of cognitive decline and dementia [7]. The evidence is clear that childhood nutrition is critical to neurological development, and malnutrition leads to long-term deficits which increase the risk of dementia later in life [8]. Other contributing clinical factors, such as inflammatory defense mechanisms associated with repeated childhood chronic illnesses, and adverse childhood experiences (ACE) are also linked to neurodegeneration and cognitive decline [9,10]. Poor living conditions and limited access to healthcare in children from lower socioeconomic backgrounds mean they are more likely to suffer repeated illnesses [11]. As socioeconomic status (SES) increases, so does access to education and stimulation. Increasingly educated parents provide more learning resources, healthier environments, and greater cognitive stimulation at home, which are linked to improved cognitive function and a reduced risk of dementia in later life [12]. Thus, we can establish that childhood health and SES are factors that pose a long-lasting impact on cognitive health and neurodevelopment.

However, establishing causative links between childhood health, childhood SES, and cognitive function in later life remains contentious. Some studies suggest that the negative effects of poor health or low SES in childhood may be mitigated by positive experiences in adulthood [13,14]. Children may reach higher educational attainment or secure more stable and engaging employment, thus mitigating some of the causative relationships [13,14]. Others oppose this, stating childhood disadvantages have long-term effects on neurodevelopmental structure and function [15], which cannot be entirely mitigated by improvements in later life [16].

In the last few decades, only one systematic review and meta-analysis on early-life factors associated with dementia and cognitive impairment in later life has been conducted [17]. However, epidemiological data in this study spanned from 1865 to 2017, perhaps lacking temporal relevance to today's society. Search strings were slightly limited for systematic and relevant evidence retrieval, and researcher funding potentially led to intrinsic bias. Thus, our systematic review evaluates current empirical evidence, published within the last decade (2014-2024). Our search strategy also included childhood health and childhood SES to ensure a more comprehensive analysis of early-life factors.

## Definition of childhood health and childhood SES

Childhood health refers to the physical, mental, and emotional well-being of children from birth through adolescence [18]. In the context of this study, childhood health is defined in various ways, ranging from general assessments to more specific measurements. Most studies

use retrospective self-reports, asking participants to rate their childhood health from excellent to poor. Additionally, childhood health can be assessed through factors like hunger or food insecurity, as well as the occurrence of illnesses, such as infections or psychological issues, during childhood.

Childhood SES refers to the economic and social conditions experienced during early life, often assessed through a combination of self-reported and objective measures [19]. Common measurements include retrospective self-assessments of social status during childhood, typically using a Likert scale (e.g., high, middle, low), as well as parental education and occupation, which are key indicators of a household's socioeconomic position. Additional variables include household income, financial stability, and community factors, such as the percentage of adults with higher education or in professional occupations. These measures are typically categorized into "high," "middle," or "low" SES and are consistent across studies, although specific variables may vary slightly.

### Research question and objectives

This systematic literature review poses the following research question: "Are there any childhood health and childhood SES factors that may predispose or be linked to cognitive impairment and dementia in later life?"

There are two objectives:

1. To systematically identify and synthesize evidence on the relationship between childhood health and SES with cognitive impairment and dementia in later life.

2. To explore the possible association between these early-life factors and the risk of cognitive impairment and dementia, thereby informing future cohort analyses and intervention strategies.

## Methods

### Design

This study adheres to the reporting standards of the Equator Network for systematic reviews [20], including the implementation of appropriate search strategies and the formulation of research questions based on the structured PICOS framework [21–23]. The eligibility and exclusion criteria for this study are presented in Table 1.

Briefly, this review focused on studies examining the impact of childhood health and SES on cognitive decline and dementia in older adults, including peer-reviewed research published in the past decade (2014–2024) to ensure the information is up to date.

**Table 1.  Eligibility and exclusion criteria using the PICOS framework.**

|  | Criteria |
| --- | --- |
| **Population** | Older adults who are at risk for or diagnosed with cognitive impairment or dementia. |
| **Interest** | The influence of childhood health and socioeconomic status (SES) on cognitive decline and dementia risk in later life. |
| **Comparison** | Comparisons between participants with diagnosed dementia or cognitive impairment and those without such diagnoses. |
| **Outcome** | The association of childhood health and SES with dementia and cognitive impairment in later life. |
| **Study Type** | Includes qualitative studies, quantitative studies, mixed methods studies, and cohort studies published in English, in peer-reviewed journals from 2014 to 2024. Excludes studies with animal subjects, government or organisational reports, books or book chapters, conference abstracts or proceedings, dissertations, theses, commentaries, editorials, and letters. |

## Search strategy

We applied the 2024 Medical Subject Headings (MeSH), Boolean operands, and truncations to source relevant studies for this systematic review. The search was conducted in three electronic databases, i.e., MEDLINE, CiNAHL, and PsycINFO, on 17th December 2024. The search strings were:

- MEDLINE: ("child* health*"[All Fields] OR "child* experience*"[All Fields] OR "child* illness"[All Fields] OR "child* disease*"[All Fields] OR "child* socioeconomic"[All Fields] OR "parental education"[All Fields] OR "parental occupation"[All Fields] OR "household income"[All Fields]) AND ("cogn* impairment"[All Fields] OR "cogn* decline"[All Fields] OR "cogn* disorder"[All Fields] OR "dementia"[All Fields] OR "Alzheimer*"[All Fields]) AND ("older adults"[All Fields] OR "older age"[All Fields] OR "older people"[All Fields] OR "elderly"[All Fields] OR "aged"[All Fields] OR "aging"[All Fields]), Filters: in the last 10 years, English.

- CiNAHL: childhood health OR childhood health issues OR childhood socioeconomic status OR (childhood adversity or childhood trauma or adverse childhood experiences) AND (cognitive impairment or cognitive dysfunction or cognitively impaired or dementia or Alzheimer) AND (older people or older adults or elderly or aged), Filter: Publication Year: 2014-2024, Peer-Reviewed, English Language, Human.

- PsycINFO: Any Field: child * health * OR Any Field: child * experience * OR Any Field: child * illness OR Any Field: child * disease * OR Any Field: child * socioeconomic OR Any Field: parental education OR Any Field: parental occupation OR Any Field: household income AND Any Field: cogn * impairment OR Any Field: cogn * decline OR Any Field: dementia OR Any Field: Alzheimer * AND Population Group: Human AND Age Group: Middle Age (40-64 yrs) OR Aged (65 yrs & older) AND Document Type: Journal Article AND Open Access AND Peer-Reviewed Journals only AND Year: 2014 To 2024.

We also conducted a manual search of the reference lists of relevant articles to identify additional studies. Following Equator guidelines, all authors meticulously reviewed the references of all included studies to eliminate duplicates and exclude articles that did not align with the scope of the study. The search was restricted to articles published in English. Duplicates were manually removed after identifying high-risk duplicate articles using the Rayyan platform. TL and AM independently assessed eligibility, and any disagreements were resolved by the final decision from AL.

## Study selection

The search strategy identified 7,616 records from electronic databases (1049 articles from MEDLINE, 6475 articles from CINAHL, and 92 articles from PsycINFO) and 24 from hand searching, resulting in 7,640 publications for initial evaluation. After removing 166 duplicates, 7,474 records were screened by title and abstract, resulting in the exclusion of 7,410 records for reasons including: not in English, was not published from 2014 to 2024, wrong study design, and irrelevant topic. Of the remaining 64 reports sought for retrieval, 2 were not retrieved, leaving 62 full-text papers to be assessed for eligibility. A further 18 reports were excluded because the outcome was not focused on exploring the impact of childhood health or childhood SES on dementia or cognitive impairment in later life. Ultimately, 44 publications were selected as the most relevant for the systematic review, comprising 18 cross-sectional studies and 26 cohort studies (Fig 1).

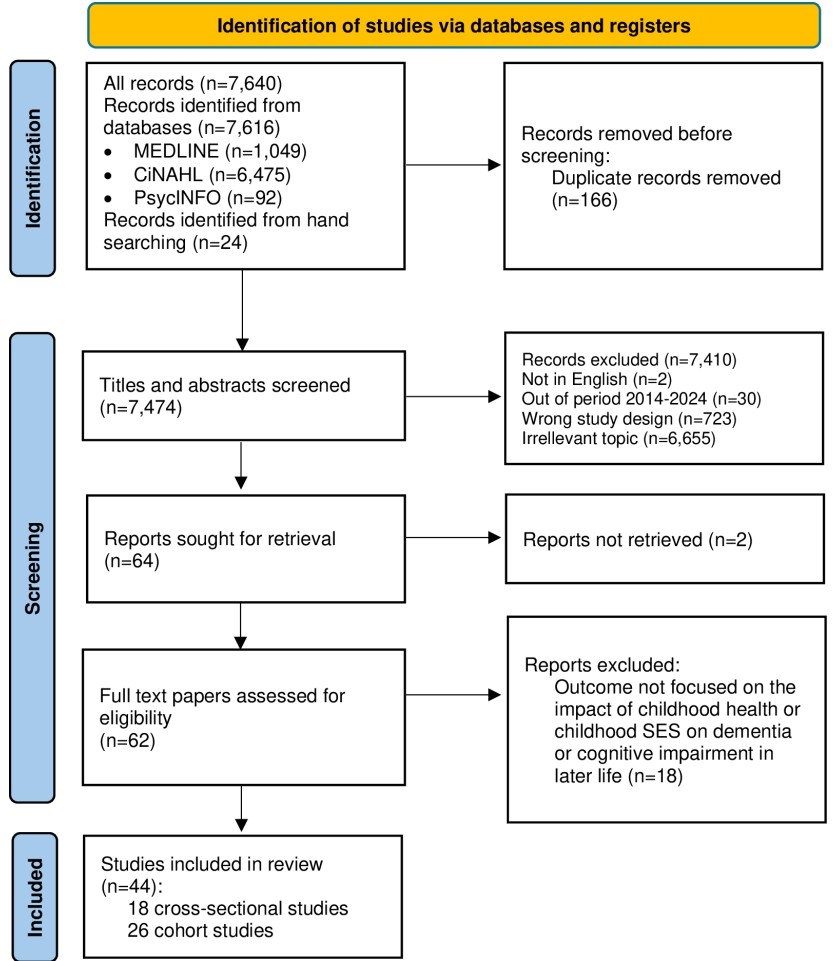

**Fig 1. PRISMA flow diagram.**

## Quality appraisal

To assess article quality for this literature review, we utilized the Joanna Briggs Institute (JBI) tool, known for its superior reliability compared to the Critical Appraisal Checklists Programme (CASP) [24]. The JBI checklist comprises eight questions for cross-sectional studies and 11 questions for cohort studies (See Tables 2 and 3 for results of quality appraisal) [25]. Responses were categorized as "yes" (1 point) and others as 0 points ("no," "unclear," or "not applicable"). Tertile classifications of high- (score > 70%), mid- (score 50–70%), and low-quality (score < 50%) were assigned using the appropriate JBI checklist. Only "high-quality" papers were included in the review. Quality assessments of each study were conducted independently by TL and AM, with cross-checking by AL to minimize bias and ensure that no data were missing in this systematic review.

## Data extraction

We extracted data from 44 articles using an Excel spreadsheet. Table 4 presents the detailed empirical research matrix: (1) Authors; (2) Title; (3) Country/Region; (4) Study design; (5) Data source; (6) Age of participants; (7) Sample size; (8) Data analysis methods; (9) Early-life factors; (10) Cognitive/dementia measures; and (11) Main outcomes.

**Table 2. JBI quality assessment for cross-sectional studies.**

| No | Authors | Title | Q1 | Q2 | Q3 | Q4 | Q5 | Q6 | Q7 | Q8 | Score (%) | Classify |
|----|---------|-------|----|----|----|----|----|----|----|----|-----------|----------|
| 1 | Lee et al. (2023) [26] | Adverse Childhood Experiences and Aging-Associated Functional Impairment in a National Sample of Older Community-Dwelling Adults | Yes | Yes | Yes | Yes | No | No | Yes | Yes | 75 | High quality |
| 2 | Brown et al. (2022) [27] | Adverse Childhood Experiences and Subjective Cognitive Decline in the US | Yes | Yes | Yes | Yes | Yes | Yes | Yes | Yes | 100 | High quality |
| 3 | Gold et al. (2021) [28] | Are adverse childhood experiences associated with late-life cognitive performance across racial/ethnic groups: results from the Kaiser Healthy Aging and Diverse Life Experiences study baseline | Yes | Yes | Yes | Yes | Yes | Unclear | Yes | Yes | 87.5 | High quality |
| 4 | Halpin et al. (2022) [29] | Are Adverse Childhood Experiences Associated with Worse Cognitive Function in Older Adults? | Yes | Yes | Yes | Yes | Yes | Unclear | Yes | Yes | 87.5 | High quality |
| 5 | Morita and Fujiwara (2021) [30] | Association between childhood parental involvement and late-life cognitive function: A population-based cross-sectional study among cognitively intact community-dwelling older adults in Japan | Yes | Yes | Yes | Yes | Yes | Yes | Yes | Yes | 100 | High quality |
| 6 | Nishizawa et al. (2019) [31] | Association between childhood socioeconomic status and subjective memory complaints among older adults: results from the Japan Gerontological Evaluation Study 2010 | Yes | Yes | Yes | Yes | Yes | Yes | Yes | Yes | 100 | High quality |
| 7 | Baiden et al. (2022) [32] | Association of adverse childhood experiences with subjective cognitive decline in adulthood: Findings from a population-based study | Yes | Yes | Yes | Yes | Yes | Unclear | Yes | Yes | 87.5 | High quality |
| 8 | Filigrana et al. (2023) [33] | Childhood and Life-Course Socioeconomic Position and Cognitive Function in the Adult Population of the Hispanic Community Health Study of Latinos | Yes | Yes | Yes | Yes | Yes | Yes | Yes | Yes | 100 | High quality |
| 9 | Kobayashi et al. (2017) [34] | Childhood deprivation and later-life cognitive function in a population-based study of older rural South Africans | Yes | Yes | Yes | Yes | Yes | Yes | Yes | Yes | 100 | High quality |
| 10 | Muhammad et al. (2022) [35] | Childhood deprivations predict late-life cognitive impairment among older adults in India | Yes | Yes | Yes | Yes | Yes | Yes | Yes | Yes | 100 | High quality |
| 11 | Rotstein and Levine (2021) [36] | Childhood infectious diseases and old age cognitive functioning: a nationally representative sample of community-dwelling older adults | Yes | Yes | Yes | Yes | Yes | Yes | Yes | Yes | 100 | High quality |
| 12 | Maharani (2019) [37] | Childhood Socioeconomic Status and Cognitive Function Later in Life: Evidence From a National Survey in Indonesia | Yes | Yes | Yes | Yes | Yes | Yes | Yes | Yes | 100 | High quality |
| 13 | Korinek et al. (2024) [38] | Cognitive function following early life war-time stress exposure in a cohort of Vietnamese older adults | Yes | Yes | Yes | Yes | Yes | No | Yes | Yes | 87.5 | High quality |
| 14 | Momtaz et al. (2015) [39] | Does food insufficiency in childhood contribute to dementia in later life? | Yes | Yes | Yes | Yes | Yes | Yes | Yes | Yes | 100 | High quality |
| 15 | Voyer et al. (2023) [40] | Linking Adverse Childhood Experiences and Other Risk Factors to Subjective Cognitive Decline in an Aging Population | Yes | Yes | Yes | Yes | Yes | Yes | Yes | Yes | 100 | High quality |
| 16 | Lian et al. (2024) [41] | No Association Found- Adverse Childhood Experiences and Cognitive Impairment in Older Australian Adults | Yes | Yes | Yes | Yes | Yes | Yes | Yes | Yes | 100 | High quality |
| 17 | Nilaweera et al. (2022) [42] | The association between adverse childhood events and later-life cognitive function and dementia risk | Yes | Yes | Yes | Yes | Yes | Yes | Yes | Yes | 100 | High quality |
| 18 | Schickedanz et al. (2022) [43] | The Association Between Adverse Childhood Experiences and Positive Dementia Screen in American Older Adults | Yes | Yes | Yes | Yes | Yes | Unclear | Yes | Yes | 87.5 | High quality |

*Q1. Were the criteria for inclusion in the sample clearly defined?; Q2. Were the study subjects and the setting described in detail?; Q3. Was the exposure measured in a valid and reliable way?; Q4. Were objective, standard criteria used for measurement of the condition?; Q5. Were confounding factors identified?; Q6. Were strategies to deal with confounding factors stated?; Q7. Were the outcomes measured in a valid and reliable way?; Q8. Was the statistical analysis used appropriate?

**Table 3. JBI quality assessment for cohort studies.**

| No | Authors | Title | Q1 | Q2 | Q3 | Q4 | Q5 | Q6 | Q7 | Q8 | Q9 | Q10 | Q11 | Score (%) | Classify |
|---|---|---|---|---|---|---|---|---|---|---|---|---|---|---|---|
| 1 | Chai et al. (2024) [44] | Association between childhood parental literacy and late-life cognitive function- The Guangzhou Biobank Cohort Study | Yes | Yes | Yes | Yes | No | Yes | Yes | Yes | Yes | No | Yes | 81.82 | High quality |
| 2 | Zhou and Zhao (2024) [45] | Childhood Peer Relationships and Dementia Risk in Chinese Older Adults- A Mediation Analysis | Yes | Yes | Yes | Yes | Yes | Yes | Yes | Yes | Yes | Yes | Yes | 100 | High quality |
| 3 | He and Yang (2024) [46] | Longitudinal association of adverse childhood experiences with cognitive function trajectories among middle-aged and older adults- group-based trajectory modeling | Yes | Yes | Yes | Yes | No | Yes | Yes | Yes | Yes | Yes | Yes | 90.91 | High quality |
| 4 | Ma et al. (2021) [47] | The influence of childhood adversities on mid to late cognitive function: From the perspective of life course | Yes | Yes | Yes | Yes | Unclear | Yes | Yes | Yes | Yes | No | Yes | 81.82 | High quality |
| 5 | Yang and Wang (2020) [48] | Early-Life Conditions and Cognitive Function in Middle-and Old-Aged Chinese Adults: A Longitudinal Study | Yes | Yes | Yes | Yes | No | Yes | Yes | Yes | Yes | Yes | Yes | 100 | High quality |
| 6 | Sha et al. (2018) [49] | Associations of childhood socioeconomic status with mid-life and late-life cognition in Chinese middle-aged and older population based on a 5-year period cohort study | Yes | Yes | Yes | No | No | Yes | Yes | Yes | Yes | No | Yes | 72.73 | High quality |
| 7 | Lin et al. (2022) [50] | Association of Adverse Childhood Experiences and Social Isolation With Later-Life Cognitive Function Among Adults in China | Yes | Yes | Yes | Yes | Yes | Yes | Yes | Yes | Yes | Yes | Yes | 100 | High quality |
| 8 | Yuan et al. (2022) [51] | Heterogeneous adverse childhood experiences and cognitive function in an elderly Chinese population: a cohort study | Yes | Yes | Yes | Yes | Yes | Yes | Yes | Yes | Yes | No | Yes | 90.91 | High quality |
| 9 | Aartsen et al., (2019) [52] | Advantaged socioeconomic conditions in childhood are associated with higher cognitive functioning but stronger cognitive decline in older age | Yes | Yes | Yes | Yes | Yes | Yes | Yes | Yes | Yes | Yes | Yes | 100 | High quality |
| 10 | Lewis et al. (2023) [53] | Availability of Cognitive Resources in Early Life Predicts Transitions Between Cognitive States in Middle and Older Adults From Europe | Yes | Yes | Yes | Yes | Unclear | Yes | Yes | Yes | Yes | Yes | Yes | 90.91 | High quality |
| 11 | Donley et al. (2018) [54] | Association of childhood stress with late-life dementia and Alzheimer's disease: the KIHD study | Yes | Yes | Yes | Yes | Yes | Yes | Yes | Yes | Yes | No | Yes | 90.91 | High quality |
| 12 | Tani et al. (2021) [55] | Adverse Childhood Experiences and Dementia: Interactions With Social Capital in the Japan Gerontological Evaluation Study Cohort | Yes | Yes | Yes | Yes | Yes | Yes | Yes | Yes | Yes | No | Yes | 90.91 | High quality |
| 13 | Tani et al. (2020) [56] | Association Between Adverse Childhood Experiences and Dementia in Older Japanese Adults | Yes | Yes | Yes | Yes | Yes | Yes | Yes | Yes | Yes | No | Yes | 90.91 | High quality |
| 14 | Gutierrez et al. (2024) [57] | My Parent, Myself, or My Child- Whose Education Matters Most for Trajectories of Cognitive Aging in Middle Age | Yes | Yes | Yes | Yes | Yes | Yes | Yes | Yes | Yes | Yes | Yes | 100 | High quality |
| 15 | Hendrie et al. (2018) [58] | The Association of Early Life Factors and Declining Incidence Rates of Dementia in an Elderly Population of African Americans | Yes | Yes | Yes | No | No | Yes | Yes | Yes | Yes | No | Yes | 72.73 | High quality |
| 16 | Racine Maurice et al. (2021) [59] | Childhood Socioeconomic Status Does Not Predict Late-Life Cognitive Decline in the 1936 Lothian Birth Cohort | Yes | Yes | Yes | Yes | No | Yes | Yes | Yes | Yes | Unclear | Yes | 81.82 | High quality |
| 17 | Dekhtyar et al. (2015) [60] | A Life-Course Study of Cognitive Reserve in Dementia--From Childhood to Old Age | Yes | Yes | Yes | Yes | Yes | Yes | Yes | Yes | Yes | No | Yes | 90.91 | High quality |
| 18 | Künzi et al. (2024) [61] | The impact of early adversity on later life health, lifestyle, and cognition | Yes | Yes | Yes | Yes | No | Yes | Yes | Yes | Yes | Yes | Yes | 90.91 | High quality |
| 19 | Tsang et al. (2022) [62] | The long arm of childhood socioeconomic deprivation on mid- to later-life cognitive trajectories: A cross-cohort analysis | Yes | Yes | Yes | Yes | Yes | Yes | Yes | Yes | Yes | Yes | Yes | 100 | High quality |

*(Continued)*

**Table 3.** (Continued)

| No | Authors | Title | Q1 | Q2 | Q3 | Q4 | Q5 | Q6 | Q7 | Q8 | Q9 | Q10 | Q11 | Score (%) | Classify |
|----|---------|-------|----|----|----|----|----|----|----|----|----|-----|-----|-----------|----------|
| 20 | Thomas et al. (2024) [63] | Early-Life Parental Affection, Social Relationships in Adulthood, and Later-Life Cognitive Function | Yes | Yes | Yes | Yes | Yes | Yes | Yes | Yes | Yes | Yes | Yes | 100 | High quality |
| 21 | Lor et al. (2023) (29) [64] | What is the association between adverse childhood experiences and late-life cognitive decline? Study of Healthy Aging in African Americans (STAR) cohort study | Yes | Yes | Yes | Yes | Yes | Yes | Yes | Yes | Yes | No | Yes | 90.91 | High quality |
| 22 | Kucharska-Newton et al. (2023) [65] | Association of Childhood and Midlife Neighborhood Socioeconomic Position With Cognitive Decline | Yes | Yes | Yes | Yes | Yes | Yes | Yes | Yes | Yes | Unclear | Yes | 90.91 | High quality |
| 23 | Greenfield et al. (2021) [66] | Life Course Pathways From Childhood Socioeconomic Status to Later-Life Cognition: Evidence From the Wisconsin Longitudinal Study | Yes | Yes | Yes | Yes | Yes | Yes | Yes | Yes | Yes | Yes | Yes | 100 | High quality |
| 24 | Lee et al. (2021) [67] | Multigenerational Households During Childhood and Trajectories of Cognitive Functioning Among U.S. Older Adults | Yes | Yes | Yes | No | No | Yes | Yes | Yes | Yes | No | Yes | 72.73 | High quality |
| 25 | Tom et al. (2020) [68] | Association of Demographic and Early-Life Socioeconomic Factors by Birth Cohort With Dementia Incidence Among US Adults Born Between 1893 and 1949 | Yes | Yes | Yes | Unclear | No | Yes | Yes | Yes | Yes | Yes | Yes | 81.82 | High quality |
| 26 | Zhang et al. (2020) [69] | Early-life socioeconomic status, adolescent cognitive ability, and cognition T in late midlife: Evidence from the Wisconsin Longitudinal Study | Yes | Yes | Yes | No | Unclear | Yes | Yes | Yes | Yes | No | Yes | 72.73 | High quality |

*Q1. Were the two groups similar and recruited from the same population?; Q2. Were the exposures measured similarly to assign people to both exposed and unexposed groups?; Q3. Was the exposure measured in a valid and reliable way?; Q4. Were confounding factors identified?; Q5. Were strategies to deal with confounding factors stated?; Q6. Were the groups/participants free of the outcome at the start of the study (or at the moment of exposure)?; Q7. Were the outcomes measured in a valid and reliable way?; Q8. Was the follow up time reported and sufficient to be long enough for outcomes to occur?; Q9. Was follow up complete, and if not, were the reasons to loss to follow up described and explored?; Q10. Were strategies to address incomplete follow up utilized?; Q11. Was appropriate statistical analysis used?

**Table 4. Empirical research matrix presenting studies included in the systematic review (N = 44).**

| No | Authors | Title | Study design | Country/ Region | Data source | Age of participants | Sample size | Data analysis methods | Early-life factors | Cognitive/ dementia measures | Main outcomes |
|---|---|---|---|---|---|---|---|---|---|---|---|
| 1 | Dekhtyar et al. (2015) [60] | A Life-Course Study of Cognitive Reserve in Dementia--From Childhood to Old Age | Cohort | Sweden | The Uppsala Birth Cohort Multigeneration Study 1915-1929 | 65+ | 7,574 | Descriptive statistics, discrete time proportional hazard models | Education | Dementia diagnosis in Medical record | The risk of dementia was lower among those with higher childhood school grades (hazard ratio [HR]: 0.79; 95% confidence interval [CI]: 0.68 to 0.93). |
| 2 | Aartsen et al. (2019) [52] | Advantaged socioeconomic conditions in childhood are associated with higher cognitive functioning but stronger cognitive decline in older age | Cohort | Europe | The Survey of Health, Ageing, and Retirement in Europe (SHARE) | 50-96 | 24,066 | Descriptive statistics, linear and non-linear mixed-effect models | Childhood SES | Delayed recall and verbal fluency | Compared to those with the most disadvantaged childhood conditions, those in the most advantaged socioeconomic positions scored 1.27 more words for delayed recall and 5.39 more words for verbal fluency at age 73. |
| 3 | Lee et al. (2023) [26] | Adverse Childhood Experiences and Aging-Associated Functional Impairment in a National Sample of Older Community-Dwelling Adults | Cross-sectional | US | The National Social Life, Health and Aging Project (NSHAP) | 50+ | 3,387 | Descriptive statistics, Chi-square tests, multivariable logistic regression models | Physical violence or abuse, witnessing of physical violence or abuse, family financial stress, separation from a parent, poor childhood health, and unhappy family life | Montreal Cognitive Assessment (MoCA-SA) | Participants were more likely to demonstrate at least mild cognitive impairment if they reported a history of poor or fair childhood health (OR 1.74, 95% CI 1.20–2.53). After adjusting for age, gender, race, and ethnicity, participants reporting any ACE history were more likely to demonstrate physical mobility impairment (OR 1.30, 95% CI 1.11–1.52) and cognitive impairment (OR 1.26, 95% CI 1.03–1.54) and report functional disability (OR 1.69, 95% CI 1.38–2.07), compared to those with no ACE history |
| 4 | Tani et al. (2021) [55] | Adverse Childhood Experiences and Dementia: Interactions With Social Capital in the Japan Gerontological Evaluation Study Cohort | Cohort | Japan | The Japan Gerontological Evaluation Study 2013-2016 | 65+ | 16,821 | Descriptive statistics, Cox regression models | Parental loss or parental divorce, family psychopathology, abuse and neglect | Mini-Mental State Examination | Those with more ACEs had a greater risk of dementia. The hazard ratio for ≥3 ACEs was 3.25 (95% CI = 1.73–6.10) among those with low social capital and 1.19 (95% CI = 0.58–2.43) among those with middle social capital. ACEs were associated with increased dementia incidence only for those with low social capital. |

*(Continued)*

**Table 4.** (Continued)

| No | Authors | Title | Country/Region | Study design | Data source | Age of participants | Sample size | Data analysis methods | Early-life factors | Cognitive/dementia measures | Main outcomes |
|----|---------|-------|----------------|--------------|-------------|---------------------|-------------|-----------------------|--------------------|------------------------------|---------------|
| 5 | Brown et al. (2022) [27] | Adverse Childhood Experiences and Subjective Cognitive Decline in the US | US | Cross-sectional | The 2019 Behavioral Risk Factor Surveillance System Survey (BRFSS) | 45+ | 82,688 | Descriptive statistics, multivariable logistic regression | ACEs included sexual, physical/psychological and environmental ACEs | Confusion and memory loss | Sexual (adjusted OR (aOR: 2.83; 95% CI: 2.42 – 3.31)), physical/psychological (aOR: 2.05; 95% CI: 1.83 – 2.29) and environmental (aOR: 1.94; 95% CI: 1.74 – 2.16) ACEs were associated with SCD. There was also a dose-response relationship between ACE score and SCD |
| 6 | Gold et al. (2021) [28] | Are adverse childhood experiences associated with late-life cognitive performance across racial/ethnic groups: results from the Kaiser Healthy Aging and Diverse Life Experiences study baseline | US | Cross-sectional | Kaiser Healthy Aging and Diverse Life Experiences Study 2017 | 65+ | 1,661 | Descriptive statistics, covariate-adjusted mixed-effects linear regression models | Parents' education, divorce, remarriage, domestic violence, witnessing substance abuse, parent's job loss, parental incarceration, family member's illness, and death of mother or father | Verbal episodic memory, semantic memory and executive functioning | Parent's remarriage (β = −0.11; 95% CI: −0.20 to −0.03), mother's death (β = −0.18; 95% CI: −0.30 to −0.07), and father's death (β = −0.11; 95% CI: −0.20 to −0.01) were associated with worse cognition. |
| 7 | Halpin et al. (2022) [29] | Are Adverse Childhood Experiences Associated with Worse Cognitive Function in Older Adults? | US | Cross-sectional | The Maine-Aging Behavior Learning Enrichment (M-ABLE) Study at the University of Maine | 55-90 | 121 | Descriptive statistics, Spearman's, Pearson, hierarchical regressions | Childhood experiences of abuse, neglect, and household dysfunction | The National Institute of Health-Toolbox for the Assessment of Neurological and Behavioral Function Quality of Life in Neurological Disorders (NIH-TB Neuro-QOL, Version 2.0) | ACE scores were negatively associated with income and years of education and positively associated with depressive symptoms and SCC. ACE scores were a significant predictor of intellectual function and executive attention; however, these relationships were no longer significant after adjusting for education |
| 8 | Tani et al. (2020) [56] | Association Between Adverse Childhood Experiences and Dementia in Older Japanese Adults | Japan | Cohort | The Japan Gerontological Evaluation Study 2013-2016 | 65+ | 17,412 | Descriptive statistics, Cox regression models | Parental loss or parental divorce, family psychopathology, and abuse and neglect | Mini-Mental State Examination | Participants who experienced 3 or more ACEs had a greater risk of developing dementia compared to those without ACEs (HR = 1.78; 95% CI: 1.15–2.75; P = 0.009). |

*(Continued)*

Table 4. (Continued)

| No | Authors | Title | Country/Region | Study design | Data source | Age of participants | Sample size | Data analysis methods | Early-life factors | Cognitive/dementia measures | Main outcomes |
|---|---|---|---|---|---|---|---|---|---|---|---|
| 9 | Morita and Fujiwara (2021) [30] | Association between childhood parental involvement and late-life cognitive function: A population-based cross-sectional study among cognitively intact community-dwelling older adults in Japan | Japan | Cross-sectional | Self-administered survey | 65–88 | 266 | Descriptive statistics, multiple regression analysis models | Parents often read books, helped with homework, discussed issues, went on outings, played outside, and cooked together | Quick Mild Cognitive Impairment | Participants with greater positive parental involvement in childhood showed higher total QMCI scores (P < 0.001). Those with low, medium, and high parental involvement had higher scores by 2.30 (95% CI: 0.36–4.24), 2.83 (95% CI: 0.41–5.23), and 6.00 (95% CI: 2.39–9.61) points, respectively, compared to those without positive parental involvement. Book reading showed a significant positive association with the total score. |
| 10 | Chai et al. (2024) [44] | Association between childhood parental literacy and late-life cognitive function-The Guangzhou Biobank Cohort Study | China | Cohort | The Guangzhou Health and Happiness Association for the Respectable Elders (GHHARE) | 50+ | 8,891 | Descriptive statistics, ANOVA and chi-square test, multivariable linear regression, logistic regression | Parental literacy | Mini-Mental State Examination and Delayed Word Recall Test | Compared with those whose childhood parents could not read/write, those with only the mother, only the father, or both parents able to read/write during childhood had higher scores on the MMSE and its dimensions, and lower odds of mild cognitive impairment (MCI) |
| 11 | Nishizawa et al. (2019) [31] | Association between childhood socioeconomic status and subjective memory complaints among older adults: results from the Japan Gerontological Evaluation Study 2010 | Japan | Cross-sectional | The Japan Gerontological Evaluation Study 2010–2011 | 65+ | 16,184 | Descriptive statistics, Poisson regression models | Childhood SES | Subjective memory complaints | Compared with those who experienced high SES during childhood, those with low SES were 1.29 times (95% CI: 1.22–1.36) and those with middle SES were 1.10 times (95% CI: 1.04–1.17) more likely to show subjective memory complaints. |

*(Continued)*

**Table 4.** (Continued)

| No | Authors | Title | Country/ Region | Study design | Data source | Age of participants | Sample size | Data analysis methods | Early-life factors | Cognitive/ dementia measures | Main outcomes |
|---|---|---|---|---|---|---|---|---|---|---|---|
| 12 | Lin et al. (2022) [50] | Association of Adverse Childhood Experiences and Social Isolation With Later-Life Cognitive Function Among Adults in China | China | Cohort | The China Health and Retirement Longitudinal Study Wave 1,2,3 2011-2015 | 45-97 | 6,466 | Descriptive statistics, linear mixed-effects models | ACEs (i.e., physical child abuse, household substance abuse, domestic violence, unsafe neighborhood, and bullying, emotional neglect, household mental illness, incarcerated household member, parental separation or divorce, and parental death) | Episodic memory and executive function | Compared with no exposures, experience of 1 deprivation-related ACE was associated with faster cognitive decline in global cognition ($\beta = -0.012$ [95% CI, $-0.022$ to $-0.002$] SD/y) and executive function ($\beta = -0.010$ [95% CI, $-0.020$ to $-0.00002$] SD/y), whereas individuals with at least 2 childhood deprivations had faster cognitive declines in all cognitive tests ($\beta = -0.035$ [95% CI, $-0.050$ to $-0.019$] SD/y for global cognition; $\beta = -0.047$ [95% CI, $-0.068$ to $-0.025$] SD/y for episodic memory; $\beta = -0.019$ [95% CI, $-0.034$ to $-0.004$] SD/y for executive function). However, such an association was not observed for threat-related ACEs. |
| 13 | Baiden et al. (2022) [32] | Association of adverse childhood experiences with subjective cognitive decline in adulthood: Findings from a population-based study | US | Cross-sectional | The 2019 Behavioral Risk Factor Surveillance System Survey (BRFSS) | 45-79 | 50,277 | Descriptive statistics, binary logistic regression | Childhood abuse (i.e., emotional, physical, and sexual) and household dysfunction (i.e., mental illness, substance use, incarceration, separation/ divorce, and witnessing domestic violence | Confusion and memory loss | Respondents who had four or more adverse childhood experiences had 2.98 times higher odds of having subjective cognitive decline when compared to respondents with no adverse childhood experiences (aOR = 2.98, 95% CI = 2.56-3.48). |

*(Continued)*

Table 4. (Continued)

| No | Authors | Title | Country/ Region | Study design | Data source | Age of participants | Sample size | Data analysis methods | Early-life factors | Cognitive/ dementia measures | Main outcomes |
|---|---|---|---|---|---|---|---|---|---|---|---|
| 14 | Kucharska-Newton et al. (2023) [65] | Association of Childhood and Midlife Neighborhood Socioeconomic Position With Cognitive Decline | US | Cohort | The community-based Atherosclerosis Risk in Communities (ARIC) Study | 45-64 | 5711 | Descriptive statistics, multinomial linear mixed-effects model | Median household income; median value of owner-occupied housing units; percentage of households receiving interest, dividend, or net rental income; percentage of adults with a high school degree; percentage of adults with a college degree; and percentage of adults in professional, managerial, or executive occupations. | Word Fluency Test, Digit Symbol Substitution Test, and Delayed Word Recall Test, Digit Span Backward Test, Boston Naming Test, Animal Naming Test, Trail Making Tests A and B, Incidental Learning Test, and Logical Memory Test | Each 1-SD-higher childhood SES score was associated with a slower (β, −9.2%; 95% CI, −12.1% to −6.4%) rate of cognitive decline relative to the sample median. A comparable association was observed when comparing the highest tertile with the lowest tertile of childhood SES (β, −17.7%; 95% CI, −24.1% to −11.3%) |
| 15 | Donley et al. (2018) [54] | Association of childhood stress with late-life dementia and Alzheimer's disease: the KIHD study | Finland | Cohort | The Kuopio Ischemic Heart Disease Risk Factor Study since 1984 | 42–61 | 2,682 | Descriptive statistics, multiple covariates analyses | Stress | ICD-8 code 290, ICD-9 codes 4378A and 290, and ICD-10 | Childhood stress associated with increased risk of dementia (HR = 1.86, 95% CI: 1.12–3.10). Associations remained statistically significant after adjustment for age, education, income, and other covariates (HR = 1.93, 95% CI: 1.14–3.25). |
| 16 | Tom et al. (2020) [68] | Association of Demographic and Early-Life Socioeconomic Factors by Birth Cohort With Dementia Incidence Among US Adults Born Between 1893 and 1949 | US | Cohort | The Adult Changes in Thought study | 65+ | 4,277 | Descriptive statistics, Fine and Gray subdistribution proportional hazards model | Childhood financial stability | The Cognitive Abilities Screening Instrument | Indicators of a more advantaged early-life environment (OR = 0.91, 95% CI 0.84-0.98) were associated with a lower incidence of dementia. Lower educational level (<= high school) (OR = 1.23, 95% CI: 1.07-1.40) were associated with a higher incidence of dementia. |

(Continued)

**Table 4.** (Continued)

| No | Authors | Title | Country/ Region | Study design | Data source | Age of participants | Sample size | Data analysis methods | Early-life factors | Cognitive/ dementia measures | Main outcomes |
|---|---|---|---|---|---|---|---|---|---|---|---|
| 17 | Sha et al. (2018) [49] | Associations of childhood socio-economic status with mid-life and late-life cognition in Chinese middle-aged and older population based on a 5-year period cohort study | China | Cohort | The China Health and Retirement Longitudi-nal Study Wave 1,2,3 2011-2015 | 45-90 | 10,533 | Descriptive statistics, latent growth curve models | Family financial status, parent's education attain-ments, health | Telephone Interview of Cognitive Status, word recall, and drawing a figure successfully | Cognition in the 45-59 age cohort showed a curvilinear change, while cognition in the 60-90 age cohort showed a linear decline. Participants with higher childhood SES, including higher self-evaluated financial status, higher parental education, and better health, were asso-ciated with better mid-life cognitive performance. |
| 18 | Lewis et al. (2023) [53] | Availability of Cog-nitive Resources in Early Life Predicts Transitions Between Cognitive States in Middle and Older Adults From Europe | Europe | Cohort | The Survey of Health, Ageing, and Retirement in Europe (SHARE) | 60+ | 32,783 | Descriptive statistics, multistate survival models, multinomial regression model | Access to books | Telephone Inter-view of Cognitive Status | Access to more books at age 10 was associated with a decreased risk of develop-ing cognitive impairment (adjusted hazard ratio = 0.79, confidence interval: 0.76–0.82). Total longevity was similar between partic-ipants reporting high (+1 standard deviation [SD]) and low (−1 SD) number of books in the childhood home; how-ever, individuals with more access to childhood books lived a greater proportion of this time without cognitive impairment. |

*(Continued)*

**Table 4.** (Continued)

| No | Authors | Title | Country/ Region | Study design | Data source | Age of participants | Sample size | Data analysis methods | Early-life factors | Cognitive/ dementia measures | Main outcomes |
|----|---------|-------|-----------------|--------------|-------------|---------------------|-------------|-----------------------|--------------------|------------------------------|---------------|
| 19 | Filigrana et al. (2023) [33] | Childhood and Life-Course Socioeconomic Position and Cognitive Function in the Adult Population of the Hispanic Community Health Study Study of Latinos | US | Cross-sectional | The Hispanic Community Health Study/ Study of Latinos (HCHS/ SOL) | 45–74 | 9,331 | Descriptive statistics, mediation analysis | Parental education | The Six-Item Screener (SIS), a brief measure of global mental status (34); the 2 scores of the Brief Spanish-English Verbal Learning Test (B-SEVLT) (35), a measure of verbal learning and memory; the Controlled Oral Word Association or Word Fluency Test (WF) (36), a measure of verbal functioning; and the Digit Symbol Substitution Test (DSS) of the Wechsler Adult Intelligence Scale–Revised, a measure of psychomotor speed and sustained attention | High childhood SEP was associated with global cognition in adulthood (coefficient for parental education beyond high school vs. less than high school = 0.26, 95% confidence interval: 0.15, 0.37). |
| 20 | Kobayashi et al. (2017) [34] | Childhood deprivation and later-life cognitive function in a population-based study of older rural South Africans | South Africa | Cross-sectional | Health and Aging in Africa: Longitudinal Study of an INDEPTH Community since 1992 | 40+ | 5,059 | Descriptive statistics, linear regression models | Health, education, parent's occupation | Time orientation, numeracy, and word recall | Poor childhood health was associated with lower cognitive scores (total effect = −0.28; 95% CI = −0.35 to −0.21, compared to good health). Having a father in a professional job during childhood, although rare (3% of the sample), was associated with better cognitive scores (total effect = 0.25; 95% CI = 0.10 to 0.40, compared to unskilled manual labor, which 29% of the sample had). Education was positively and linearly associated with later-life cognitive function (effect = 0.09; 95% CI = 0.09 to 0.10 per additional year of education). |

*(Continued)*

**Table 4.** (Continued)

| No | Authors | Title | Country/ Region | Study design | Data source | Age of participants | Sample size | Data analysis methods | Early-life factors | Cognitive/ dementia measures | Main outcomes |
|---|---|---|---|---|---|---|---|---|---|---|---|
| 21 | Muhammad et al. (2022) [35] | Childhood deprivations predict late-life cognitive impairment among older adults in India | India | Cross-sectional | The Longitudinal Ageing Study in India (LASI) Wave 1 2017-2018 | 60+ | 31,464 | Descriptive statistics, moderated multiple linear regression models | Childhood health status (good, fair and poor) and childhood SES (good, average, poor) | Mini-Mental State Examination | Older adults who had fair health during childhood were more likely to suffer from cognitive impairment compared to those with good childhood health (Coef: 0.60; CI 0.39–0.81). Those with poor childhood financial status were more likely to suffer from cognitive impairment compared to those with good childhood financial status (Coef: 0.81; CI 0.56–1.07). Older adults with both fair childhood health and poor childhood financial status were more likely to suffer from cognitive impairment compared to those with both good childhood health and good financial status (Coef: 1.26; CI 0.86–1.66). |
| 22 | Rotstein and Levine (2021) [36] | Childhood infectious diseases and old age cognitive functioning: a nationally representative sample of community-dwelling older adults | Ireland | Cross-sectional | The Irish Longitudinal Study on Ageing (TILDA) | 65-85 | 2,994 | Descriptive statistics, linear regression models | Childhood infectious diseases (i.e., chicken pox, measles, and mumps) | Mini-Mental State Examination | The most parsimonious model was a linear adjusted model (Bayesian Information Criterion = 12646.09). Late-life cognitive functioning significantly improved as the number of childhood infectious diseases increased (β = 0.18; 95% CI = 0.11, 0.26; p < 0.001). This effect was not significantly attenuated in all sensitivity analyses. |

*(Continued)*

Table 4. (Continued)

| No | Authors | Title | Country/Region | Study design | Data source | Age of participants | Sample size | Data analysis methods | Early-life factors | Cognitive/dementia measures | Main outcomes |
|---|---|---|---|---|---|---|---|---|---|---|---|
| 23 | Zhou and Zhao (2024) [45] | Childhood Peer Relationships and Dementia Risk in Chinese Older Adults- A Mediation Analysis | China | Cohort | The China Health and Retirement Longitudinal Study Wave 1,2,3,4 2011-2018 | 60+ | 7,574 | Descriptive statistics, marginal structural models | Childhood Peer Relationships | Telephone Interview for Cognitive Status and its modifications (TICS-m) 10-word immediate and delayed recall test for episodic memory (score range: 0–20), a serial 7's subtraction task for working memory (score range: 0–5), and a backwards counting task for attention (score range: 0–2). | Individuals with deficits in childhood peer relationships had a higher risk of dementia (odds ratio [OR], 1.21; 95% confidence interval [CI], 1.10–1.34) compared with those with more positive experiences. |
| 24 | Racine Maurice et al. (2021) [59] | Childhood Socioeconomic Status Does Not Predict Late-Life Cognitive Decline in the 1936 Lothian Birth Cohort | Scotland | Cohort | The Lothian Birth Cohort 1936 (LBC1936) | 70-82 | 519 | Descriptive statistics, multiple linear regressions | Parental Socio-demographics, Parental Social Class | Mini-Mental State Examination | Participants with less educated mothers showed an increase in cognitive decline (β = −0.132, p = 0.048, and CI = −0.80, −0.00). The relationship between maternal educational attainment and cognitive decline became non-significant when controlling for adult SES (i.e., participant educational attainment and occupation) |
| 25 | Maharani (2019) [37] | Childhood Socioeconomic Status and Cognitive Function Later in Life: Evidence From a National Survey in Indonesia | Indonesia | Cross-sectional | The Indonesia Family Life Survey (IFLS) Wave 5 2014-2015 | 50+ | 6,676 | Descriptive statistics, linear regression models | Hunger, the availability of facilities, and the number of books in the home | Telephone Interview for Cognitive Status | Numbers of facilities and books available in childhood homes substantially associated with cognition in later life. |
| 26 | Korinek et al. (2024) [38] | Cognitive function following early life war-time stress exposure in a cohort of Vietnamese older adults | Vietnam | Cross-sectional | The Vietnam Health and Aging Study | 59+ | 2,447 | Descriptive statistics, quantile regression | Severe hunger in childhood and environmental hardships, war stress exposure | Mini-Mental State Examination | Severe childhood hunger and environmental hardships are linked to poorer cognitive function in later life. PTSD, hypertension, and stroke, all exacerbated by wartime stress, are also associated with lower cognitive scores. |

(Continued)

Table 4. (Continued)

| No | Authors | Title | Country/ Region | Study design | Data source | Age of participants | Sample size | Data analysis methods | Early-life factors | Cognitive/ dementia measures | Main outcomes |
|---|---|---|---|---|---|---|---|---|---|---|---|
| 27 | Momtaz et al. (2015) [39] | Does food insufficiency in childhood contribute to dementia in later life? | Malaysia | Cross-sectional | The Mental Health and Quality of Life of Older Malaysians 2003-2005 | 60+ | 2,745 | Descriptive statistics, multiple binary logistic regression | Food insufficiency | Geriatric Mental State-Automated Geriatric Examination for Computer Assisted Taxonomy | Food insufficiency in childhood independently increased the risk of developing dementia in old age by 81%, after adjusting for socio-demographic factors (odds ratio = 1.81, 95% confidence interval 1.13–2.92, P < 0.01). |
| 28 | Yang and Wang (2020) [48] | Early-Life Conditions and Cognitive Function in Middle-and Old-Aged Chinese Adults: A Longitudinal Study | China | Cohort | The China Health and Retirement Longitudinal Study Wave 1,2,3 2011–2015 | 45-101 | 16,258 | Descriptive statistics, multilevel growth curve modeling | Early parental death, childhood SES (i.e., education and occupation of father), food deprivation, and childhood health | Episodic memory and mental intactness | Early maternal death was linked to lower episodic memory scores at baseline, with a decrease of 0.20 points compared to those without this experience. In contrast, better childhood factors such as enhanced paternal education, a non-agricultural job, food security, and good childhood health were associated with better episodic memory. Higher childhood socioeconomic status (SES) predicted higher baseline cognition in both age groups, though it only protected against cognitive decline at baseline in middle-aged adults. |
| 29 | Thomas et al. (2024) [63] | Early-Life Parental Affection, Social Relationships in Adulthood, and Later-Life Cognitive Function | US | Cohort | Three waves of the Mid-life in the United States (MIDUS) study, telephone interviews and self-administered questionnaires | 50+ | 1,983 | Descriptive statistics, structural equation modeling (SEM) | Parental Affection in Childhood | Immediate word list recall, delayed word list recall, backward digit span, number series, counting backward speed task, and category fluency | Significant indirect effects of parental affection on better cognitive function through higher levels of social support (both average social support and family social support) in adulthood in the full sample. |
| 30 | Zhang et al. (2020) [69] | Early-life socio-economic status, adolescent cognitive ability, and cognition in late midlife: Evidence from the Wisconsin Longitudinal Study | US | Cohort | The Wisconsin Longitudinal Study Wave 1957, 1964, 1975, 1993, 2004, and 2011 | 65+ | 5,880 | Descriptive statistics, structural equation models | Parent's education, parent's occupation, childhood household income, education | Immediate and delayed recall, digit ordering, letter and category fluency, and a subset of the Wechsler Adult Intelligence Scale similarities test. | Childhood SES had a direct effect on cognition in late midlife (β = 0.072, p < 0.01). Mother's education (CFI = 0.499), father's education (CFI = 0.719), father's occupational education (CFI = 0.682), household income (CFI = 453). |

*(Continued)*

**Table 4.** (Continued)

| No | Authors | Title | Country/ Region | Study design | Data source | Age of participants | Sample size | Data analysis methods | Early-life factors | Cognitive/ dementia measures | Main outcomes |
|---|---|---|---|---|---|---|---|---|---|---|---|
| 31 | Yuan et al. (2022) [51] | Heterogeneous adverse childhood experiences and cognitive function in an elderly Chinese population: a cohort study | China | Cohort | The China Health and Retirement Longitudinal Study Wave 1,2,3,4 2011-2018 | 60+ | 7,222 | Descriptive statistics, χ2 test, rank-sum test, binary logistic models | Abuse (physical and emotional), neglect (physical and emotional), household dysfunction (substance abuse, incarceration, mental illness, violence, parental separation or absence), and living surroundings (bullying and community violence) | Mini-Mental State Examination | Child maltreatment was related to a higher risk of cognitive impairment (OR = 1.37, 95% CI: 1.12 to 1.68), compared with low ACEs participants. |
| 32 | Greenfield et al. (2021) [66] | Life Course Pathways From Childhood Socioeconomic Status to Later-Life Cognition: Evidence From the Wisconsin Longitudinal Study | US | Cohort | The Wisconsin Longitudinal Study Wave 1957, 1964, 1975, 1993, 2004, and 2011 | 72 | 3,706 | Descriptive statistics, structural equation modeling (SEM), Confirmatory factor analysis | Childhood SES | Verbal fluency, working memory, immediate and delayed word recall test (episodic memory) | Scholastic performance in adolescence and midlife status attainment together fully mediated associations between childhood SES and both memory and language/ executive functioning at age 72. Adolescent scholastic performance was directly associated with later-life cognition, as well as indirectly through midlife status attainment |

*(Continued)*

Table 4. (Continued)

| No | Authors | Title | Country/ Region | Study design | Data source | Age of participants | Sample size | Data analysis methods | Early-life factors | Cognitive/ dementia measures | Main outcomes |
|----|---------|-------|-----------------|--------------|-------------|---------------------|-------------|-----------------------|--------------------|------------------------------|---------------|
| 33 | Voyer et al. (2023) [40] | Linking Adverse Childhood Experiences and Other Risk Factors to Subjective Cognitive Decline in an Aging Population | US | Cross-sectional | The 2019 Behavioral Risk Factor Surveillance System Survey (BRFSS) | 45+ | 17,042 | Descriptive statistics, multivariate logistic regression | ACEs included sexual, physical/ psychological and environmental ACEs | Survey respondents were asked if they had "experienced confusion or memory loss that is happening more often or is getting worse" in the past 12 months (15). If a participant responded affirmatively, they were identified as having SCD and were then asked a series of 5 additional questions regarding their level of difficulty with day-to-day activities, whether they needed help with these activities, whether they were able to get help when needed, whether SCD interfered with socialization, and whether they had discussed their confusion or memory loss with a clinician | Two or more ACEs also significantly increased the odds of SCD (AOR, 1.69; 95% CI, 1.36–2.10). |
| 34 | He and Yang (2024) [46] | Longitudinal association of adverse childhood experiences with cognitive function trajectories among middle-aged and older adults- group-based trajectory modeling | China | Cohort | The China Health and Retirement Longitudinal Study Wave 1,2,3,4 2011–2018 | 45+ | 1,679 | Descriptive statistics, group-based Trajectory Modelling, multinomial unordered logistic models | Child maltreatment, exposure to violence, parent/ sibling death or disability, and parental maladjustment | The assessment of episodic memory encompasses both immediate recall and delayed recall. The assessment of mental intactness includes time orientation, overlapping pentagon drawing, and arithmetic | Three cognitive decline subgroups emerged: low-start decline, high- start stability, and mid-start decline. There is no dose-response relationship between cumulative adverse childhood experiences and cognitive function. |

*(Continued)*

Table 4. (Continued)

| No | Authors | Title | Country/Region | Study design | Data source | Age of participants | Sample size | Data analysis methods | Early-life factors | Cognitive/dementia measures | Main outcomes |
|---|---|---|---|---|---|---|---|---|---|---|---|
| 35 | Lee et al. (2021) [67] | Multigenerational Households During Childhood and Trajectories of Cognitive Functioning Among U.S. Older Adults | US | Cohort | The Health and Retirement Study (1998–2014) | 51+ | 8,799 | Descriptive statistics, growth curve models | Childhood family structure | Telephone Interview of Cognitive Status | Childhood family structure was significantly linked to cognitive functioning. Individuals from multigenerational households, including those living with a single parent and grandparents, showed higher cognitive functioning compared to those from two-parent households. |
| 36 | Gutierrez et al. (2024) [57] | My Parent, Myself, or My Child- Whose Education Matters Most for Trajectories of Cognitive Aging in Middle Age | Mexico | Cohort | The Mexican Health and Aging Study (MHAS) | 50+ | 8,822 | Descriptive statistics, linear mixed models | Parental education, education | The Cross-Cultural Cognitive Examination | Lower educational levels in both parents ($\beta = -0.005$) and respondents ($\beta = -0.013$) were linked to faster decline in delayed verbal memory scores, but not in immediate verbal memory scores. |
| 37 | Lian et al. (2024) [41] | No Association Found- Adverse Childhood Experiences and Cognitive Impairment in Older Australian Adults | Australia | Cross-sectional | The Personality and Total Health (PATH) Through Life Project Wave 4 | 72-79 | 1,568 | Descriptive statistics, multiple logistic regressions | Childhood SES | Dementia diagnosis in Medical record | No significant association between childhood adversity and the presence of cognitive impairment or dementia across all tested models. |
| 38 | Nilaweera et al. (2022) [42] | The association between adverse childhood events and later-life cognitive function and dementia risk | France | Cross-sectional | The Etude Santé Psychologique Prévalence Risques et Traitement (ESPRIT) 1999–2001 | 65+ | 1,562 | Descriptive statistics, logistic regression models, Cox proportional | Abuse or maltreatment, death of a parent, prolonged parental hospitalization, serious illness of a parent or child, poverty and financial difficulties, strict upbringing, war or natural disaster, and frequent conflicts at home. | Diagnostic and Statistical Manual of Mental Disorders, Mini-Mental State Examination, Benton's Visual Retention Test for visual memory, Isaacs Set Test for verbal fluency | Individuals with multiple adverse childhood events had an increased risk of poor psychomotor speed at baseline, with those experiencing 3-4 events having an odds ratio (OR) of 1.39 and those with ≥5 events having an OR of 1.52. Worse verbal fluency was observed in individuals with 3-4 ACEs (OR: 1.34). Early-life abuse/maltreatment (OR: 1.47) and poverty/financial difficulties (OR: 1.53) were associated with worse psychomotor speed. |

(Continued)

Table 4. (Continued)

| No | Authors | Title | Country/ Region | Study design | Data source | Age of participants | Sample size | Data analysis methods | Early-life factors | Cognitive/ dementia measures | Main outcomes |
|----|---------|-------|-----------------|--------------|-------------|---------------------|-------------|------------------------|--------------------|------------------------------|---------------|
| 39 | Schickedanz et al. (2022) [43] | The Association Between Adverse Childhood Experiences and Positive Dementia Screen in American Older Adults | US | Cross-sectional | The 2017 wave of the Panel Study of Income Dynamics (PSID) and the 2014 PSID Childhood Retrospective Circumstances Survey | 65+ | 1,488 | Descriptive statistics, binomial logistic analytic model, adjusted ordered logistic regression | Parent mental illness, parent substance abuse, parent intimate partner violence, parental divorce or separation, deceased or absent parent, physical abuse, sexual abuse, emotional abuse, and neglect experienced | The AD8 dementia screen, a judgment, orientation, and function participant or informant-reported, 8-item measure of worsening cognitive impairment used to discriminate dementia from normal cognition. | Older adults with ≥4 ACEs had higher rates of a positive dementia screen (AD8 score ≥2 points) compared to those with no ACEs (26.6% vs. 16.3%, p = 0.034). Respondents with ≥4 ACEs also had higher odds of a 1-point increase in AD8 score across all intervals (adjusted odds ratio: 1.79, 95% CI: 1.05–3.04). |
| 40 | Hendrie et al. (2018) [58] | The Association of Early Life Factors and Declining Incidence Rates of Dementia in an Elderly Population of African Americans | Nigeria | Cohort | The Indianapolis-Ibadan project 1992–2009 | 70+ | 3,276 | Descriptive statistics, t-tests, Fisher's exact tests, Cox proportional hazards regression models | Education | The community screening interview for dementia (CSID) | Significant interaction (p = 0.0477) between education and childhood for AD risk. Higher education level associated with reduced AD risk (HR = 0.87). |
| 41 | Künzi et al. (2024) [61] | The impact of early adversity on later life health, lifestyle, and cognition | UK | Cohort | The English Longitudinal Study of Ageing (ELSA) and the UK Biobank | 40-73 | 515,013 | Descriptive statistics, path analysis, The Full Information Maximum Likelihood (FIML) estimation method | Physical assault, sexual assault, parental abuse, deprivation, physical neglect, sexual abuse, emotional neglect, physical abuse, emotional abuse | Immediate memory recall, visual declarative memory | Parental abuse was associated with poorer immediate memory and verbal fluency. Deprivation at age 10 predicted worse memory and verbal fluency. Physical neglect in childhood was associated with more errors in visual memory tasks. |
| 42 | Ma et al. (2021) [47] | The influence of childhood adversities on mid to late cognitive function: From the perspective of life course | China | Cohort | The China Health and Retirement Longitudinal Study Wave 2,3 2013-2015 | 45+ | 9,942 | Descriptive statistics, structural equation models | Parents' education, father's occupation, self-assessed household economic status, lack of friends, parental mental health problems, and parent–child relationships | Telephone Interview of Cognitive Status | Parental mental health problems during childhood and poor parent–child relationships both significantly influenced cognitive decline (β = -0.190, SE = 0.011; β = -0.033, SE = 0.012). |

*(Continued)*

**Table 4.** (Continued)

| No | Authors | Title | Country/ Region | Study design | Data source | Age of participants | Sample size | Data analysis methods | Early-life factors | Cognitive/ dementia measures | Main outcomes |
|----|---------|-------|-----------------|--------------|-------------|---------------------|-------------|-----------------------|--------------------|------------------------------|---------------|
| 43 | Tsang et al. (2022) [62] | The long arm of childhood socioeconomic deprivation on mid- to later-life cognitive trajectories: A cross-cohort analysis | UK, US | Cohort | The Whitehall II study, the Health and Retirement Study, and the Kame Project | 50+ | 15,309 | Descriptive statistics, latent class mixed models, logistic regressions | Parental education, parental unemployment, and family financial hardship | Mini-Mental State Examination, verbal memory word free recall, written naming task of words beginning with the letter "S", the Telephone Interview for Cognitive Status | Lower childhood SES was generally associated with a faster cognitive decline trajectory, placing individuals in lower cognitive trajectory classes. |
| 44 | Lor et al. (2023) [64] | What is the association between adverse childhood experiences and late-life cognitive decline? Study of Healthy Aging in African Americans (STAR) cohort study | US | Cohort | The Study of Healthy Aging in African Americans 2018-2019 | 50+ | 764 | Descriptive statistics, linear mixed models | Parents' divorce or separation, a parent remarrying, witnessing domestic violence, substance abuse by a family member, parental job loss, parental incarceration, serious illness of a family member, and the death of a mother or father. | Executive function (constructed from category fluency, phonemic/letter fluency and working memory) and verbal episodic memory | Compared with no ACEs, two ACEs (β=0.117; 95%CI 0.052 to 0.182), three ACEs (β=0.075; 95% CI 0.007 to 0.143) and four or more ACEs (β=0.089; 95% CI 0.002 to 0.158) were associated with less decline in executive function |

*aOR: Adjusted Odds Ratio; ACE: Adverse Childhood Experiences; AD8: Alzheimer's Disease 8 (screening test); β: Beta (coefficient); CI: Confidence Interval; CFI: Comparative Fit Index; DSM-IV: Diagnostic and Statistical Manual of Mental Disorders, 4th Edition; HR: Hazard Ratio; MCI: Mild Cognitive Impairment; MMSE: Mini-Mental State Examination; OR: Odds Ratio; P: P-value; QMCI: Quantitative Measure of Cognitive Impairment; SCD: Subjective Cognitive Decline; SCC: Subjective Cognitive Complaints; SES: Socioeconomic Status; SEP: Socioeconomic Position; SD: Standard Deviation; SDS: Social Dominance Scale; TMTB: Trail Making Test B.

# Results

## Study characteristics

Research on the relationship between childhood health, SES, and later-life dementia spans globally. This review identified a notable focus on resource-rich countries. The US published 15 studies [26–29,32,33,40,43,63–69], plus a multicentre study with the UK [62]. China published eight studies [44–51], and Japan published four studies [30,31,55,56]. Two studies were conducted across Europe [52,53], while two others were conducted in the UK [61], including one in Scotland [59]. The remaining countries–Australia [41], Finland [54], France [42], India [35], Indonesia [37], Ireland [36], Malaysia [39], Mexico [57], Nigeria [58], South Africa [34], Sweden [60], and Vietnam [38]–each have only one study. This highlights a diverse but uneven distribution of research.

Age ranges varied across participants, but the baseline was at least 40 years old because all studies aimed to evaluate any links between childhood health and childhood SES and cognitive function later in life. Sample sizes across studies varied significantly, ranging from 121 [29] to 515,013 [61], with the risk that the smaller population studies may lack sufficient power to detect meaningful differences or associations.

A number of population databases were noted across the 44 studies. Sixteen studies analyzed data from the Health and Retirement Studies International Family of Studies, including the Health and Retirement Study (HRS) [62,67], the China Health and Retirement Longitudinal Study (CHARLS) [45–51], the Indonesia Family Life Survey (IFLS) [37], the English Longitudinal Study of Aging (ELSA) [61], the Survey of Health, Aging, and Retirement in Europe (SHARE) [52,53], the Mexican Health and Aging Study (MHAS) [57], the Irish Longitudinal Study on Ageing (TILDA) [36], and the Longitudinal Ageing Study in India (LASI) [35]. Using these validated population datasets offers several strengths, such as harmonized data collection, diverse cross-cultural samples, and longitudinal assessments of cognitive function. Longitudinal data tracking cognitive changes over time allows for in-depth analyses of the long-term impact of childhood experiences. However, a notable limitation of this methodology is reliance on retrospective self-reports of childhood health and SES and associated recall bias.

Other sources of data across the studies were retrieved from surveys, such as the Behavioral Risk Factor Surveillance System Survey (BRFSS) [27,32,40], the Japan Gerontological Evaluation Study (JAGES) [31,55,56], and the Wisconsin Longitudinal Study (WLS) [66,69]. Each survey drew from prevalidated cognitive assessment tools, such as dementia diagnosis, medical records, and MMSE. These tools are highly reliable in identifying dementia-related outcomes. Papers drawn from survey data had smaller study populations when compared with larger longitudinal datasets. This may reduce the generalizability and cross-cultural applicability of the findings, but they offer contextual, country-specific foci.

In summary, compared to other studies, those in the HRS family excel at capturing long-term, cross-cultural trends, though they still encounter challenges related to recall bias.

A number of pre-validated measurement instruments were applied across the evidence. For cognitive function and dementia diagnosis, most studies utilized the Mini-Mental State Examination (MMSE) [35,36,38,42,44,51,55,56,59]. MMSE is a validated tool for assessing cognitive function in community settings due to its quick administration, ease of use, and robust validation across diverse populations [70]. The MMSE's 11 items evaluate five domains of cognitive function: orientation, registration, attention, recall, and language, providing a comprehensive overview of an individual's cognitive status. This makes it particularly suitable for large-scale surveys and effective screening of cognitive impairment in older populations [70]. Other tools included the Cognitive Abilities Screening Instrument (CASI) [68], Community Screening Interview for Dementia (CSI 'D') [58], Cross-Cultural Cognitive Examination

(CCCE) [57], Delayed Word Recall Test (DWRT) [44,63,65,66], Geriatric Mental State (GMS) [39], ICD-8, 9, & 10 [54], medical records [41,60], Quick Mild Cognitive Impairment (QMCI) [30], Telephone Interview for Cognitive Status (TICS) [37,45,62], Montreal Cognitive Assessment (MoCA-SA) [26], and Wechsler Adult Intelligence Scale (WAIS) [69].

Variables related to childhood health, childhood SES, ACEs, parental engagement, education, nutritional status, living conditions, and books read were included, ensuring a comprehensive evaluation of the link between childhood health, childhood SES, and cognitive impairment or dementia in later life.

Only 11 papers reported response rates [26,30,31,34,36,40,48,51,55–57]. The highest response rate (91.8%) was observed in the MHAS from the HRS family of surveys [57], whereas the lowest response rate (35.3%) was in a Japanese cross-sectional study [30]. This is notable from a methodological perspective, as higher response rates may be subject to response and selection bias. Individuals with better cognitive function, higher SES, or greater engagement with healthcare systems are more likely to participate, therefore overrepresenting the healthier populations. Recognizing these biases is essential when interpreting results, particularly in studies evaluating datasets that measure early-life factors against later-stage cognitive outcomes. Moreover, most studies relied on retrospective information about childhood health and childhood SES, which may be subject to recall bias and/or missing data. To mitigate these biases, a range of advanced statistical techniques such as structural equation modeling (SEM) [63,66], latent growth curve modeling [49], sensitivity analysis [36], and path analysis [61] were used. To determine linear relationships, binary, multinomial, and multivariate regression models were applied [26,28,31,34–37,42,43,55,56,58]. Mixed-effects models evaluated temporal trends [28,50,65]. More advanced regression techniques were employed to assess relative effects. These included Cox proportional hazards models [42,55,56,58], multistate survival models [53], and group trajectory models [46]. More specific data distributions and between-group comparisons across variables were statistically evaluated using appropriate tests such as chi-square [26,44], ANOVA [44], and Fisher's exact test [58]. Additionally, quantile regression [38] and marginal structural models [45] were applied to address specific data distributions and explore causal relationships.

## The long arm of childhood circumstances on cognitive function and dementia

A review of all data and information across studies revealed five common themes, presenting factors describing the associations between childhood health and childhood SES, and potential links with dementia and cognitive function. These include: (1) childhood health; (2) childhood educational attainment; (3) family socioeconomic and educational factors; (4) childhood experiences; and (5) childhood reading habits and social interactions.

**Childhood health.** Several studies identified links between childhood health and cognitive impairment or dementia later in life. Sha et al.'s longitudinal study of 10,533 participants linked better childhood health with 1.1 times higher cognitive performance in middle and older age when compared with those reporting poorer childhood health [49]. Cross-sectional studies report even higher likelihood estimates of 1.4–1.74 across India [35] and the US [26]. Similarly, Kobayashi et al.'s population-based analysis of older people linked poor childhood health history to a 28% reduction in cognitive scores later in life [34]. Donley et al.'s Finnish longitudinal study of 2,682 participants revealed that childhood stress approximately doubled the risk of dementia [54]. Findings related to childhood nutrition were also noteworthy. Momtaz's (2015) Malaysian cross-sectional study of 2,745 participants revealed that childhood food scarcity nearly doubled the likelihood of cognitive impairment in old age [39]. This

finding was slightly higher than those reported in studies conducted in Vietnam [38] and China [48], which linked severe childhood hunger to a 1.5 times increased risk of poor cognitive outcomes. Conversely, analysis of data from the Irish longitudinal study on aging linked improved cognitive functions with childhood infectious diseases. They found the more infectious diseases experienced by children, the better their cognitive function later in life. This study of 2,994 participants posits that each additional infectious disease may be associated with a 0.18-point improvement in cognitive functioning [36].

**Childhood educational attainment.** Childhood education has also been significantly linked to later-life cognitive function. Koyabishi et al.'s study found that each additional year of education may increase cognitive function scores in later life by 0.09 points [34]. Similarly, Hendrie et al.'s (2018) Nigerian study of 3,276 participants found that each additional year of education reduced the risk of dementia by 7% [58]. These figures are reflected in studies across the US (1.23 times higher risk of dementia with lower education attainment) [68]. In Sweeden, Dekhtyar et al.'s (2015) retrospective analysis of 7,574 students, aggregated to subject-specific performance, found links between lowest grade point average students and a 1.21 times higher risk of dementia in later life [60].

**Family socioeconomic and educational factors.** This systematic review also indicates a significant body of evidence linking family socioeconomic and education factors with dementia. Studies across India, the US, the UK, and China revealed individuals in midlife or older who had reported experiencing financial hardship during childhood tend to exhibit greater cognitive decline, ranging from 9% to 22% when compared to those with more favorable financial conditions [35,48,49,68]. A slightly higher figure was reported by a cross-study in Indonesia, which found that lack of basic amenities such as electricity, running water, and indoor toilets during childhood was associated with lower cognitive function scores later in life, with a difference of up to 26% [37]. Another cross-sectional study in South Africa found that individuals whose fathers held professional jobs had cognitive scores 0.25 points higher than those whose fathers worked in unskilled manual labor [34]. Additionally, a study conducted in Japan reported that compared to individuals with high SES during childhood, those with low SES and middle SES were 1.29 times and 1.1 times more likely to report subjective memory complaints in later life, respectively [31]. A retrospective study using data from the WLS found that an increase of one standard deviation in childhood SES was associated with memory scores higher by 8% and language/executive function scores higher by 34% at age 72 [66]. Aartsen et al.'s (2019) European longitudinal study of 24,066 participants [52], compared lower childhood SES (disadvantaged) populations with higher SES (advantaged) populations and identified significant reductions in verbal fluency and recall by age 73 in the most disadvantaged groups. A US cross-sectional study [33] of 9,331 participants aged 45-74 linked parental higher educational status with a 0.26 higher average score for their children on cognitive tests in later life. Children of parents who are literate and educated were found to have 0.5 times higher memory scores when compared to lower-educated parents [44].

However, Racine et al.'s Scottish cohort analysis [59] found that a lower education status mother may predict an increased risk of cognitive decline, yet the study indicated additional confounding factors. They found father's employment as manual labor, or reduced educational attainment, had no impact on cognitive outcome measures.

**Childhood experiences.** The childhood experiences summarized in this review encompass various factors influencing children's development. These experiences are categorized into key groups: abuse and neglect, including physical, emotional, and sexual abuse, lack of parental care and attention or absence of peer relationships; family-related issues, such as parental loss, divorce, remarriage, domestic violence, or parental mental health problems and substance abuse; and adverse living environments including unsafe neighborhoods.

There is a wealth of evidence supporting links between high levels of ACEs and increased risk of dementia. Longitudinal studies in Japan [55,56] and the US [32], indicate individuals with three or more ACEs had a 1.78 to 3.25 times higher risk of developing dementia compared to those without such experiences. A French study estimated that individuals with three to four ACEs had a 1.39 times higher risk of reduced psychomotor speed, and risks increased with each additional ACE [42]. Those with five or more ACEs had a 1.52 times higher risk compared to individuals with two or fewer ACEs [42]. Lor et al's study of healthy aging in Africa, with 764 participants linked to two ACEs posed an 11.7% higher risk of cognitive decline, those with three ACEs had a 7.5% higher risk, and those with four or more ACEs had an 8.9% higher risk, perhaps indicating ACES is not cumulative [64].

Moving on from comparative studies determining how many ACES cumulatively affect cognitive functioning in later life, we can assert that there is a link. This is supported by a large cross-sectional US study (N = 1,488), which identified a 10.3% higher prevalence of dementia in older adults with four or more ACEs [43].

Childhood sexual abuse is also associated with a 1.37 increased risk of cognitive decline in later life [51]. Another US-based study found this even higher, citing that individuals who experienced sexual abuse as a child had a 2.83 times greater likelihood of subjective cognitive decline when compared to those without [27].

Parent-child relationships are also highly important, with one linking medium to high levels of positive parental involvement during childhood with increased cognitive score outcomes [30]. A large UK longitudinal evaluation of 515,013 participants revealed the significant impact of parental abuse on future cognitive development, memory, verbal fluency, and recall [61]. Other negative family events, (such as parental remarriage, and parent death) have been linked to lower cognitive outcomes in later life. Specifically, parental remarriage was associated with a 0.11-point decrease in cognitive test scores, maternal death with a 0.18-point decrease, and paternal death with a 0.11-point decrease [28]. Conversely, an Australian study of 1568 participants revealed no links between ACEs and diagnoses of dementia [41].

**Childhood reading habits and social interactions.** There is also evidence supporting childhood reading habits and social interactions and potential links with cognitive function. A Japanese study found regular reading resulted in 3.11 higher test scores in later life when compared to non-readers [30]. Individuals with more books in the household had higher cognitive outcome scores and a 21% risk reduction of dementia [37,53]. Furthermore, longitudinal studies in China found that individuals who faced difficulties in establishing peer relationships during childhood scored 0.208 points lower on cognitive tests and had a 21% higher risk of developing dementia compared to those with positive peer relationships [45,47].

## Discussion

This study highlights poor childhood health, low educational attainment, adverse socioeconomic conditions, ACEs, limited reading habits, and inadequate social interactions increase the risk of dementia later in life. It could be explained that chronic illnesses and malnutrition during childhood disrupt neurodevelopment [71], negatively affecting brain structure and function. Such disruptions result in reduced grey matter volume and alterations in white matter integrity, which are known contributors to long-term cognitive decline [72–74]. However, one study reported that children who experienced multiple infectious diseases during childhood scored higher on cognitive assessments [36]. This counterintuitive finding suggests a need for further research to understand potential mechanisms linking childhood immune challenges and cognitive outcomes.

Several studies showed that higher education levels are consistently associated with better cognitive function. This aligns with prior evidence suggesting that higher education levels were linked to slower brain function decline, lower prevalence of cognitive disorders [75,76], and better short- and long-term memory performance throughout life [77]. Childhood SES also plays a pivotal role in later cognitive outcomes. Children from low-income families face barriers to adequate nutrition, healthcare, educational opportunities, and limited access to intellectually stimulating activities [78–82]. Moreover, substandard living conditions—such as homes lacking electricity or running water—exacerbate childhood stress and impair development [37,83,84]. These factors hinder optimal brain development and predispose children to poorer cognitive function [85]. Furthermore, a stable family structure and higher parental education levels significantly contribute to cognitive function later in life [86]. This can be explained by the fact that parents with higher education levels, leading to stable and well-paying jobs, are better positioned to provide resources, opportunities, academic support, and an environment conducive to cognitive development [87–89].

ACEs, including abuse, neglect, and family dysfunction, have consistently been associated with poorer cognitive outcomes. These findings corroborate previous evidence suggesting that stress and trauma during critical periods of brain development lead to structural and functional changes in the brain [90–92]. Such alterations often affect regions involved in stress regulation and cognitive functioning, resulting in long-term impairments. Additionally, the presence of both parents in a family, compared to single-parent families or those without parental presence, has been linked to better emotional regulation and social skills, which are crucial for cognitive health [93]. However, one cross-sectional study conducted in Australia did not find an association between ACEs and cognitive impairment [41]. This finding may be explained by the fact that the study only included participants from Canberra (the capital) and Queanbeyan, which are both relatively affluent areas [94]. These cities have a higher concentration of wealth, and individuals residing in such environments may have experienced lower levels of childhood adversity compared to populations from less advantaged regions. As a result, this could have influenced the study's findings.

A notable finding in this study is that access to books during childhood is associated with a reduced risk of dementia. While evidence on this topic is limited in previous research, this finding can be explained by the cognitive stimulation provided by reading, which enhances language skills and promotes lifelong learning—factors crucial for maintaining cognitive resilience [95]. Children engaged in reading activities are more likely to develop critical thinking and comprehension skills, contributing to improved educational and cognitive outcomes [95]. However, as previously noted, limited research has explored the influence of childhood reading on cognitive function in later life, emphasizing the need for further studies to provide a clearer understanding of its role in cognitive development.

Finally, the importance of social interactions during childhood was also recorded. This finding can be explained by the role of social interactions in developing key skills such as communication, emotional regulation, and problem-solving [96]. Children who experience social isolation are deprived of opportunities to cultivate these skills, which can adversely impact cognitive health [97].

This study synthesizes global evidence, emphasizing the importance of early-life interventions in preventing cognitive decline. Public health initiatives should prioritize enhancing children's physical and mental health while addressing socioeconomic inequalities. These strategies not only benefit individuals but also alleviate the societal and economic burden of cognitive decline and dementia, aligning with WHO guidelines [4]. Beyond the scope of these guidelines, this study shows that promoting reading habits from an early age may serve as a potential protective factor for cognitive health, supporting existing theories on cognitive

reserve. Future research is needed to clarify the causal relationship between reading habits and cognitive development. Additionally, a deeper exploration of the impact of childhood infectious diseases on later-life cognition is warranted to inform future preventive strategies.

This study may have some limitations. Firstly, it was restricted to English-language papers published between 2014 and 2024, potentially excluding relevant research published in other languages or from earlier periods. Secondly, a substantial proportion of the studies were cross-sectional, which limits the ability to establish causality between risk factors and dementia. Thirdly, the variation in sample sizes among the included studies could affect the reliability of the results. There were also inconsistencies in the criteria used to define exposure factors in some studies may impact the comparability and validity of the findings and, as many studies drew from the same datasets, (such as CHARLS, WLS, and JAGES), and thus, results may be skewed through duplicate counting.

## Conclusions and implications

Childhood health, childhood educational attainment, family socioeconomic and educational factors, childhood experiences, and childhood reading habits and social interactions were all linked to the risk of cognitive impairment and dementia in later life. To establish causal links between early-life risk factors and later cognitive impairment, further superior longitudinal studies are essential. Policymakers should prioritize early childhood development programs that combine health, nutrition, education, and social support to mitigate the incidence of dementia and cognitive impairment in later life. Over 60% of studies included in this review concentrated in a few wealthy nations (the US, China, and Japan), likely reflecting disparities in research capacity and funding priorities at the global level. It also raises concerns about whether the findings, which specific socioeconomic and cultural contexts may influence, can be generalized to other parts of the world, particularly those with fewer resources and different social structures. Future research should prioritize broader geographic inclusion to ensure a more comprehensive and equitable understanding of how childhood experiences influence dementia risk.

## Supporting information

**S1 Appendix. PRISMA checklist.**
(DOCX)

**S2 Appendix. Systematic review screening process.**
(XLSX)

## Author contributions

**Conceptualization:** Mark Hayter, Amanda Lee.

**Data curation:** Tung Le.

**Formal analysis:** Tung Le.

**Methodology:** Amanda Lee.

**Project administration:** Tung Le.

**Supervision:** Asri Maharani, Amanda Lee.

**Validation:** Asri Maharani.

**Writing – original draft:** Tung Le.

**Writing – review & editing:** Asri Maharani, Mark Hayter, James Gilleen, Amanda Lee.

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
