## [Decision Letter · Decision Letter 0]

29 Oct 2024

PONE-D-24-40492Cognitive impairment and dementia – Are they linked to childhood health and socioeconomic status? A systematic reviewPLOS ONE

Dear Dr. Le,

Thank you for submitting your manuscript to PLOS ONE. After careful consideration, we feel that it has merit but does not fully meet PLOS ONE’s publication criteria as it currently stands. Therefore, we invite you to submit a revised version of the manuscript that addresses the points raised during the review process.

As the reviewers point out, the manuscript in its current version should be better contextualized, especially by including the most recent publications on the subject. This aspect would substantially improve the relevance of the contribution of your research. In addition, some reviewers indicate in their comments to me the need to improve the writing style of your paper. Therefore, I strongly recommend you to have a professional English style review by a native speaker.

We look forward to receiving your revised manuscript.

Kind regards,

Jordi Gumà, Ph.D.

Academic Editor

PLOS ONE

Journal Requirements:

2. As required by our policy on Data Availability, please ensure your manuscript or supplementary information includes the following: 

3. Please note that your Data Availability Statement is currently missing the DOI/accession number of each dataset and a direct link to access each database. If your manuscript is accepted for publication, you will be asked to provide these details on a very short timeline. We therefore suggest that you provide this information now, though we will not hold up the peer review process if you are unable.

Reviewers' comments:

Reviewer's Responses to Questions

**Comments to the Author**

1. Is the manuscript technically sound, and do the data support the conclusions?

Reviewer #1: Yes

Reviewer #2: Partly

Reviewer #3: No

2. Has the statistical analysis been performed appropriately and rigorously? 

Reviewer #1: I Don't Know

Reviewer #2: N/A

Reviewer #3: N/A

3. Have the authors made all data underlying the findings in their manuscript fully available?

Reviewer #1: Yes

Reviewer #2: Yes

Reviewer #3: No

4. Is the manuscript presented in an intelligible fashion and written in standard English?

Reviewer #1: Yes

Reviewer #2: Yes

Reviewer #3: Yes

5. Review Comments to the Author

Reviewer #1: General comment

GC#1: Thank you for the opportunity to read this interesting article. The topic addressed by the authors of this systematic review is of great interest, and therefore, an article aimed at synthesizing the scientific evidence on the possible association between the onset of cognitive decline and childhood health conditions and socioeconomic status is of utmost cultural and scientific relevance.

Introduction

I#1: At line 46, after the introductory section and before addressing the topic of the role of “early-life nutrition” (line 49), it would be appropriate to guide the reader gradually towards the factors investigated in this systematic review. This could be done by introducing the concept of ‘risk factor’ as a useful element in addressing a health issue of such significant impact in terms of burden on individuals, the population, and healthcare systems, especially considering the aging population and the resulting importance of taking preventive action.

I#2: In the first part of the article, it might be appropriate to include epidemiological information and data on the frequency associated with the factors investigated, which would be useful for framing and sizing the phenomenon.

I#3: It would be useful to contextualize the work conducted by placing it within the framework of risk factor characterization and in relation to prevention. In recent years, there has been a significant production of scientific evidence regarding risk factors for dementia and cognitive decline. Specifically, to date, 14 known risk factors for dementia have been identified (12 at the time indicated in the manuscript), starting from early life (such as low education). In addition, prevention strategies to counter the risk stemming from exposure to these risk factors have been identified and described, including in internationally relevant documents and WHO guidelines. Although the central topic of this review does not concern the known risk factors for dementia, including a paragraph that describes the literature and reference documents on this subject would provide a way to relate the work of this systematic review to what is happening in the field of dementia prevention. This topic could be revisited in the discussion, emphasizing the general need to act preventively from the earliest stages of life, in line with population-wide interventions.

I#4: In describing the purpose of the systematic review, the objectives through which the authors aim to achieve it could be presented more precisely. Specifically, in the first objective, the factors on which the search string was built could be listed in greater detail. In the second objective, it might be preferable to refer to the 'possible association between the factors described in the first objective and the risk of cognitive decline' rather than 'potential factors.' In any case, this comment is merely a suggestion to make the reading clearer and to facilitate the transition to the following methods section, where the search string is provided.

Methods

M#1: As currently presented, the search string does not allow for proper replicability of the research. It would be advisable to provide the exact search string as it was launched in the databases.

M#2: Just before the figure related to the eligibility and exclusion criteria, a sentence summarizing the criteria adopted could be included to facilitate reading.

Results

R#1: In the flowchart shown in figure 1, related to the identification procedure, the number of studies identified for each database should be provided, in addition to the total number of records. The same should be done in the text, just before the figure. Additionally, the authors could specify the type of the 19 studies identified for data extraction.

R#2: The number of records obtained seems, perhaps, too limited (likely due to the particularly rigid search string). This led me to wonder whether the authors had considered more comprehensive alternatives to the search string used, in order to include a larger number of records and avoid potential selection bias.

R#3: In the section on main results (lines 196–199), it would be appropriate to provide more detailed and complete information on the study's findings, particularly regarding confounding factors. In other words, this section could be more interesting if developed in a more exhaustive manner, becoming a general introduction to the following sections; otherwise, it could be removed.

R#4: The sections "Incidence of dementia" and "Cognitive function scores" are very interesting. However, as currently presented, they do not seem to have any connection with the objective of this review. For this reason, in relation to the following section, it would be appropriate to present the data on dementia incidence (as done by the authors) while also including the main information from the studies cited in this section. For example, the objectives of the cited studies could be provided, along with details about the populations studied (clearly referring to the table where data extracted from the analyzed studies is reported).

The following sentence serves as an example to better explain this suggestion: the study “…”, whose objective was “…”, reported an incidence of “…” in a population of subjects with “…”.

Discussion

D#1: The discussion presents a brief list of considerations on the results collected in this systematic review. This section, in itself, seems more like an extension of the results rather than a proper discussion. The variables examined in this systematic review are extremely interesting and of great relevance to the scientific community focused on the prevention of cognitive decline and dementia. In my opinion, it would be important to further develop the discussion, aiming to create a link with current scientific evidence, guidelines, and international documents on dementia prevention, risk factors, and the need to address these through population-level strategies that also take into account the factors investigated in this work. In summary, the authors should discuss the public health implications of this study, as it contributes to the existing body of evidence, suggesting further elements for debate on risk factors, social determinants of health, and strategies to be defined and implemented. This concept is rightly touched upon in the conclusions but should be developed starting from the discussion.

Figures and tables

FT#1: Although the acronyms are mentioned in the text, it would be advisable to spell them out in the table (particularly in Table 4) by including the acronym and its corresponding meaning at the end of the table.

FT#2: In Table 2, it would be appropriate to include the name of the first author and the year, in addition to the title of the article. The score obtained from each study could, alternatively, be included in Table 4 by adding a column. This would make the table self-explanatory regarding the quality of the studies as well.

FT#3: The flow diagram figure appears to be of low graphical quality and, for this reason, is very difficult to read. In general, as reported in the comment R#1. The number of studies identified for each database should be provided, in addition to the total number of records.

Reviewer #2: Thank you for the opportunity to review this interesting systematic review of early life factors related to cognitive impairment. Here, the authors reviewed three large databases to identify 1,505 records and eventually include 19 journal articles linking childhood health and SES to dementia and/or cognitive impairment. While I believe the line of inquiry is timely and important, the paper in its current form suffers from important limitations that I believe if the authors can address, will be more likely to contribute meaningfully to this area of literature.

Please see below for my detailed comments that I hope will help to improve the clarity and impact of the manuscript.

Major comments:

1. Search is out of date (completed one year ago) – consider rerunning the search to see if any additional papers would be captured.

2. I was initially surprised that only 19 papers were included given the broad exposure topic of childhood health and SES. I believe this is because the only search term for ‘childhood soceioeconomic* may miss important variables related to childhood health, including those you have extracted from these papers, such as childhood ACEs, parental involvement, parental income or occupation, childhood reading habits and social interactions, and education. By not explicitly searching the databases for these factors, but rather only extracting them from the studies that been labelled ‘socioeconomic’ factors, you may be missing key areas of literature that may bias your assessment and synthesis presented here. Consider either a) rerunning your search to include as many of these terms as possible; b) re-writing your methods to be clear of your pragmatism in this regard; c) remove childhood SES as a factor of interest for this review.

3. It is noted that data from the same studies were included in the review via multiple journal articles (i.e. CHARLS, JAGES) please consider including this as a limitation and explain how it may bias the results of the review.

4. There is a lack of explanation on what ‘childhood health’ in this context entails. Please explain in more detail what measurements were used, what variables they included, and whether they were similar across studies.

Minor comments:

Introduction

- Paragraph 2: “the bulk of research” – very vague, consider revising.

- Paragraph 2: a rationale/link between childhood illness and SES would be useful and improve rationale.

- Please include quality indicators (e.g. low, moderate, high) for the JBI tool and what threshold would be used to exclude articles from synthesis.

Methods

- The methods and results sections overlap a bit too much. For instance, the data extraction should be in the methods. Consider revising for clarity and conciseness.

Results

- Figure 1 blurry and hard to read.

- Records excluded (even at title and abstract stage) should include a reason for exclusion.

- Consider including a rationale for why 2013 was chosen as the start date.

- Tables 2 and 3: please include the questions or a legend to the questions (top row) for quality assessment. At present it is unclear what the studies have been scored against. Please also include references for each of the studies in the first column.

- Paragraph 2: Consider explaining how reliability was calculated and what it means in this context.

- Paragraph 2: This section is very vague.. consider revising:

“While acknowledging potential confounders, many articles offer strategies to address biases, such as handling missing data and using analytical models. Despite focusing on childhood experiences, these factors remain relevant for future investigations.

o I’m not sure what this means.

“Importantly, data collection occurred pre-Covid-19, ensuring its integrity”.

o Integrity with regards to what?

“Studies utilised meticulously crafted questionnaires distributed through national surveys in multiple countries, ensuring rigorous testing”.

o What does rigor constitute in this context?

- The paragraph ‘main findings’ is confusing, please revise for clarity or consider removing.

- The results sections on incidence of dementia and cognitive function scores are detailing findings that were not a clear objective of the study. Consider adding this objective or removing these results.

Discussion:

- Include broader implications in discussion. What value added does this study bring?

Reviewer #3: This manuscript presented a systematic review of studies examining surveys of aging and dementia versus childhood trauma and low SES. Mixed reviews correlations have been found in prior work. Thus the goal of this paper was to determine the strength of evidence in relation to different early childhood factors. Systematic review of several databases followed by handsearching and removing duplicates revealed 19 sources that examined the research questions.

This review, if original to this field, could represent an important contribution. I am unsure whether there are other related or similar reviews, as this was not mentioned (though it should be stated explicitly). The analysis is presented topically (types of studies, study quality, demographics/tools, dementia incidence, cognitive function, childhood health, educational attainment, SES, childhood 'experiences', reading and 'social habits'). While the facts about these sources are laid out, there seems to be a lack of critical analysis of the sources or synthesis of information. The problems begin in this area. While there is a section called quality analysis, there does not appear to be any discussion of study quality in the manuscript. Bald face facts about study characteristics are listed, but not in an interesting manner as they could be. For example, authors note the countries that the data comes from, but there is not commentary on the regions that appear or not in the data. Further, there is no explanation of the survey tools used in the reviewed studies. Are readers assumed to understand these? I'm not familiar myself, but I would have found this type of source useful should this type of info been explained. One of the most incriminating omissions is lack of commentary about the types of questions or sample sizes in these tools. For example, authors note that there were >31k respondents to one survey but not which survey. A more important question would be which surveys got higher response rates and what types of respondent biases these represent, which is quite an important issue in survey data. There is no direct comparison of the type of data presented in sources, aside from the listing of data from these sources. Further, within the results section, it starts to become unclear which of the sources are represented in the types of data for specific subsections. There is also inconsistent formatting in source description, citation and statistics in these subsections of the results.

For improvement of this paper, I would like to see the authors synthesis and analyze the sources critically. What are the strengths and weaknesses of each source individually, plus in comparison? Which sources seem to show evidence against childhood effects or even weaker effects? There are a few smaller issues, but these issues seem fairly substantial so significant work seems needed for improvement.

6. PLOS authors have the option to publish the peer review history of their article (what does this mean? ). If published, this will include your full peer review and any attached files.

**Do you want your identity to be public for this peer review?** For information about this choice, including consent withdrawal, please see our Privacy Policy .

Reviewer #1: No

Reviewer #2: No

Reviewer #3: No

---

## [Author Response · Author response to Decision Letter 1]

6 Feb 2025

Manchester, 20th January 2025

To: Dr Jordi Gumà, Academic Editor, PLOS ONE

Re: Revised Manuscript (PONE-D-24-40492)

Dear Dr Gumà,

We thank you for your interest in our submission to the PLOS ONE journal and the opportunity to revise and resubmit our manuscript. We appreciate the extensive comments provided following the peer review, each of which we have now considered carefully. In response, we have undertaken a substantial revision of our manuscript in line with the issues and suggestions raised by you and the three reviewers. For clarification, we provide a point-by-point response to these comments below.

We sincerely hope our manuscript now meets your approval for publication in PLOS ONE.

On behalf of the Authors,

Yours sincerely,

Tung Le

EDITOR COMMENTS

Comments from Editor

Thank you for submitting your manuscript to PLOS ONE. After careful consideration, we feel that it has merit but does not fully meet PLOS ONE’s publication criteria as it currently stands. Therefore, we invite you to submit a revised version of the manuscript that addresses the points raised during the review process.

As the reviewers point out, the manuscript in its current version should be better contextualized, especially by including the most recent publications on the subject. This aspect would substantially improve the relevance of the contribution of your research.

Authors’ response

Thank you for highlighting this important comment from the reviewers. In response, we have revised the manuscript by incorporating the latest research on the topic. We re-conducted our literature search on December 17, 2024, to ensure we included the most recent publications. Additionally, based on the reviewers' recommendations, we expanded the search terms related to childhood socioeconomic status (SES) to include median household income, parental education level, and parental occupational status to identify more relevant articles for this systematic review.

Comments from Editor

In addition, some reviewers indicate in their comments to me the need to improve the writing style of your paper. Therefore, I strongly recommend you to have a professional English style review by a native speaker.

Authors’ response

We gratefully acknowledge the suggestions to improve the writing style of our manuscript. In response, we have carefully reviewed and refined the text to ensure clarity, coherence, and polished language. Additionally, the final manuscript was thoroughly reviewed by Dr James Gilleen and Dr Amanda Lee, native English speakers in our research group, to enhance its readability and quality further.

JOURNAL REQUIREMENTS

Journal Requirements

Authors’ response

We have ensured that our manuscript complies with the journal's formatting guidelines, including file naming conventions. The manuscript has been formatted by the PLOS ONE style templates provided at the links, ensuring alignment with the specified requirements for the main body, title, authors, and affiliations.

Journal Requirements

2. As required by our policy on Data Availability, please ensure your manuscript or supplementary information includes the following:

A numbered table of all studies identified in the literature search, including those that were excluded from the analyses. For every excluded study, the table should list the reason(s) for exclusion.

Authors’ response

We have ensured full compliance with the journal's Data Availability policy and have provided all necessary data in the manuscript and supplementary materials. Specifically:

• A table listing all studies identified in the literature search, including those excluded from the analysis with reasons for exclusion, is included in the supporting information file.

• None of the included studies are unpublished, and all primary research sources are publicly available.

• The data extracted from primary study sources for the systematic review are provided in the supporting file, detailing all relevant information required to replicate our analyses. These details include names of data extractors and the data extraction date; all extracted data used for the systematic review; and sources and dates for any additional data obtained via correspondence with original study authors or other sources.

• Tables 2 and 3 in the manuscript present the Joanna Briggs Institute (JBI) quality assessment, which includes the completed risk of bias and quality/certainty assessments for each study and confirm that studies are eligible for inclusion in the review.

• A detailed explanation of how missing data were addressed is included in the “Methods, Quality appraisal” section of the manuscript.

• All relevant information has been provided in the main text, supplementary materials, or the manuscript itself in accordance with the journal's requirements.

Journal Requirements

3. Please note that your Data Availability Statement is currently missing the DOI/accession number of each dataset and a direct link to access each database. If your manuscript is accepted for publication, you will be asked to provide these details on a very short timeline. We therefore suggest that you provide this information now, though we will not hold up the peer review process if you are unable.

Authors’ response

We have included each dataset's DOI/accession number and direct links to access each database in the supporting file.

Journal Requirements

Authors’ response

We have included the captions for our Supporting Information files and updated any in-text citations to match accordingly. 

REVIEWER 1 COMMENTS

Comments from Reviewer #1

Thank you for the opportunity to read this interesting article. The topic addressed by the authors of this systematic review is of great interest, and therefore, an article aimed at synthesizing the scientific evidence on the possible association between the onset of cognitive decline and childhood health conditions and socioeconomic status is of utmost cultural and scientific relevance.

At line 46, after the introductory section and before addressing the topic of the role of “early-life nutrition” (line 49), it would be appropriate to guide the reader gradually towards the factors investigated in this systematic review. This could be done by introducing the concept of ‘risk factor’ as a useful element in addressing a health issue of such significant impact in terms of burden on individuals, the population, and healthcare systems, especially considering the aging population and the resulting importance of taking preventive action.

Authors’ response

Thank you for the helpful suggestion. In response, we have revised the manuscript to better guide the reader towards the factors investigated in this systematic review. We introduced the concept of 'risk factor' after the introductory section, before discussing early-life nutrition (Introduction, lines 47-50):

“Consequently, understanding causation is crucial to underpin targeted early interventions and diagnoses. It can inform policy and healthcare strategies that may mitigate the significant burden of dementia, and reinforce the disease as a public health priority.”

Comments from Reviewer #1

In the first part of the article, it might be appropriate to include epidemiological information and data on the frequency associated with the factors investigated, which would be useful for framing and sizing the phenomenon.

Authors’ response

Thank you for the suggestion. We have incorporated relevant epidemiological information and data on the frequency associated with the factors investigated (Introduction, lines 60-66):

“Malnutrition affects over 149 million children under five and has significant long-term neurodevelopmental consequences [5]. Over 356 million children are living in extreme poverty and lacking access to basic healthcare, food, education, and resources [6]. There is evidence linking poor health or adverse socioeconomic conditions in childhood with later-stage cognitive decline, suggesting those from poorer backgrounds had a 1.5 to 2.0 times higher risk of cognitive decline and dementia [7].”

Comments from Reviewer #1

It would be useful to contextualize the work conducted by placing it within the framework of risk factor characterization and in relation to prevention. In recent years, there has been a significant production of scientific evidence regarding risk factors for dementia and cognitive decline. Specifically, to date, 14 known risk factors for dementia have been identified (12 at the time indicated in the manuscript), starting from early life (such as low education). In addition, prevention strategies to counter the risk stemming from exposure to these risk factors have been identified and described, including in internationally relevant documents and WHO guidelines. Although the central topic of this review does not concern the known risk factors for dementia, including a paragraph that describes the literature and reference documents on this subject would provide a way to relate the work of this systematic review to what is happening in the field of dementia prevention. This topic could be revisited in the discussion, emphasizing the general need to act preventively from the earliest stages of life, in line with population-wide interventions.

Authors’ response

Thank you for the suggestion. We have included scientific evidence related to risk factors for dementia and cognitive impairment, along with references to WHO guidelines, to contextualize our work within the broader framework of dementia prevention (Introduction, lines 51-59).

“The Lancet Commission on Dementia Prevention, Intervention, and Care identified 14 modifiable risk factors, including childhood-related factors such as low education, which account for 40% of global dementia cases [3]. Lifecourse preventative strategies are supported by the World Health Organization (WHO) 2019 report, “Reducing the Risk of Cognitive Decline and Dementia” [4]. Their guidelines highlight the impact of early-life interventions in mitigating cognitive decline and reinforce a need to evaluate childhood health and socioeconomic conditions as potential contributors to later-life cognitive decline. Thus, this systematic review seeks to reveal early-life risk factors linked to cognitive impairment and dementia.”

Comments from Reviewer #1

In describing the purpose of the systematic review, the objectives through which the authors aim to achieve it could be presented more precisely. Specifically, in the first objective, the factors on which the search string was built could be listed in greater detail. In the second objective, it might be preferable to refer to the 'possible association between the factors described in the first objective and the risk of cognitive decline' rather than 'potential factors.' In any case, this comment is merely a suggestion to make the reading clearer and to facilitate the transition to the following methods section, where the search string is provided.

Authors’ response

Thank you for your suggestion. We fully agree with you and have revised the research objectives to provide greater clarity and alignment with our search strategy and analysis approach (Introduction, lines 119-124):

“ 1. To systematically identify and synthesize evidence on the relationship between childhood health and SES with cognitive impairment and dementia in later life.

2. To explore the possible association between these early-life factors and the risk of cognitive impairment and dementia, thereby informing future cohort analyses and intervention strategies.”

Comments from Reviewer #1

As currently presented, the search string does not allow for proper replicability of the research. It would be advisable to provide the exact search string as it was launched in the databases.

Authors’ response

Thank you for your feedback. We have now included the exact search string as it was launched in the databases to ensure proper replicability of the research (Methods, Search strategy, lines 144- 167).

“• MEDLINE: ("child* health*"[All Fields] OR "child* experience*"[All Fields] OR "child* illness"[All Fields] OR "child* disease*"[All Fields] OR "child* socioeconomic"[All Fields] OR "parental education"[All Fields] OR "parental occupation"[All Fields] OR "household income"[All Fields]) AND ("cogn* impairment"[All Fields] OR "cogn* decline"[All Fields] OR "cogn* disorder"[All Fields] OR "dementia"[All Fields] OR "Alzheimer*"[All Fields]) AND ("older adults"[All Fields] OR "older age"[All Fields] OR "older people"[All Fields] OR "elderly"[All Fields] OR "aged"[All Fields] OR "aging"[All Fields]), Filters: in the last 10 years, English.

• CiNAHL: childhood health OR childhood health issues OR childhood socioeconomic status OR ( childhood adversity or childhood trauma or adverse childhood experiences ) AND ( cognitive impairment or cognitive dysfunction or cognitively impaired or dementia or Alzheimer) AND ( older people or older adults or elderly or aged ), Filter: Publication Year: 2014-2024, Peer-Reviewed, English Language, Human.

• PsycINFO: Any Field: child* health* OR Any Field: child* experience* OR Any Field: child* illness OR Any Field: child* disease* OR Any Field: child* socioeconomic OR Any Field: parental education OR Any Field: parental occupation OR Any Field: household income AND Any Field: cogn* impairment OR Any Field: cogn* decline OR Any Field: dementia OR Any Field: Alzheimer* AND Population Group: Human AND Age Group: Middle Age (40-64 yrs) OR Aged (65 yrs & older) AND Document Type: Journal Article AND Open Access AND Peer-Reviewed Journals only AND Year: 2014 To 2024.”

Comments from Reviewer #1

Just before the figure related to the eligibility and exclusion criteria, a sentence summarizing the criteria adopted could be included to facilitate reading.

Authors’ response

Thank you for your suggestion

---

## [Editor Report · Decision Letter 1]

11 Feb 2025

Cognitive impairment and dementia – Are they linked to childhood health and socioeconomic status? A systematic review

PONE-D-24-40492R1

Dear Dr. Le,

We’re pleased to inform you that your manuscript has been judged scientifically suitable for publication and will be formally accepted for publication once it meets all outstanding technical requirements.

Kind regards,

Jordi Gumà, Ph.D.

Academic Editor

PLOS ONE

---

## [Editor Report · Acceptance letter]

PONE-D-24-40492R1

PLOS ONE

Dear Dr. Le,

I'm pleased to inform you that your manuscript has been deemed suitable for publication in PLOS ONE. Congratulations! Your manuscript is now being handed over to our production team.

Kind regards,

on behalf of

Dr. Jordi Gumà

Academic Editor

PLOS ONE